# Passive eDNA collection enhances aquatic biodiversity analysis

Cindy Bessey [1,2,3 ✉], Simon Neil Jarman [3,4], Tiffany Simpson[5], Haylea Miller[2], Todd Stewart[6], John Kenneth Keesing [1,3] & Oliver Berry [2]

Environmental DNA (eDNA) metabarcoding is a sensitive and widely used approach for species detection and biodiversity assessment. The most common eDNA collection method in aquatic systems is actively filtering water through a membrane, which is time consuming and requires specialized equipment. Ecological studies investigating species abundance or distribution often require more samples than can be practically collected with current filtration methods. Here we demonstrate how eDNA can be passively collected in both tropical and temperate marine systems by directly submerging filter membranes (positively charged nylon and non-charged cellulose ester) in the water column. Using a universal fish metabarcoding assay, we show that passive eDNA collection can detect fish as effectively as active eDNA filtration methods in temperate systems and can also provide similar estimates of total fish biodiversity. Furthermore, passive eDNA collection enables greater levels of biological sampling, which increases the range of ecological questions that eDNA metabarcoding can address.

[1] Commonwealth Scientific and Industrial Research Organisation, Indian Oceans Marine Research Centre, Oceans and Atmosphere, Crawley, WA, Australia. [2] Commonwealth Scientific and Industrial Research Organization, Indian Oceans Marine Research Centre, Environomics Future Science Platform, Crawley, WA, Australia. [3] UWA Oceans Institute, University of Western Australia, Crawley, WA, Australia. [4] School of Biological Sciences and the UWA Oceans Institute, University of Western Australia, Crawley, WA, Australia. [5] eDNA Frontiers, Trace and Environmental DNA Laboratory, School of Molecular and Life Sciences, Curtin University, Perth, WA, Australia. [6] Bass Marine Pty Ltd, Port Denison, WA, Australia. ✉email: Cindy.Bessey@csiro.au

Environmental DNA (eDNA) metabarcoding is a sensitive and broadly applicable tool for biodiversity research and environmental monitoring that is noninvasive and promises to be both cost- and time-effective[1]. Macroorganisms can shed their DNA into the air, soil and water through many means, including faeces, sloughed tissue cells or as gametes, which can be collected as intra- or extracellular particles[2,3]. These DNA particles are then extracted, amplified using primer sets designed to target specific taxa, sequenced and compared to a reference database of known DNA sequences to determine taxonomic identities[4]. eDNA metabarcoding removes the need for multiple taxonomic experts in processing survey samples and allows for the simultaneous identification of various eukaryotic species from diverse trophic levels[5]. The exponential increase of published eDNA metabarcoding studies is evidence of the usefulness of this tool in addressing biological questions both on land and in the water[6,7].

eDNA has been most extensively collected in aquatic environments to investigate the presence and diversity of many species, most frequently fishes, primarily by filtering water through a membrane[6,8–11]. Although centrifugation and precipitation can be used to concentrate eDNA from small water volumes[12], filtration is more often used because it can accommodate a larger water volume that improves detection of rare species[11]. Filtering strategies vary from collecting water for filtration in a laboratory setting using a vacuum or peristaltic pump[9,13] to in situ field sampling using enclosed filters and pumps or syringes to force water through the membrane[14,15]. Despite the variability in strategies, the use of a pumping mechanism to achieve active filtration is typical[10].

Although active filtration enables a predetermined volume of water to be used for consistent sampling, it is time-consuming and requires pumping equipment[16]. The volume of water needed to sufficiently describe the biological community depends on the species diversity within the system[17], the eDNA shedding rates of each species, as well as the combined environmental factors affecting eDNA retention, resuspension, persistence, decay and transportation[18]. For example, acidic environments are known to accelerate the degradation of eDNA[19], while cool, flowing water (i.e. river) systems may transport eDNA up to 10 km from the source[20]. Although some studies indicate that sampling small water volumes (such as 1 L) can adequately detect macroorganisms of interest in some systems[21], other studies suggest at least 20 L of water per site, or more, must be sampled before species accumulation curves approach an asymptote for diversity[9,22,23]. It is not only time-consuming to pass such a large volume of water through membranes, especially when employing filters with small pore sizes, but it is often impossible, as membranes can become blocked by particulates. Researchers are now exploring alternative approaches, such as using in situ remotely deployed sampling instruments that automate filtering[24], high-volume sampling using tow nets[25] or by recovering eDNA from the tissues of marine sponges, which naturally filter thousands of litres in a day[26]. These alternative approaches remain in development, require additional equipment or are expensive[24], while organisms that naturally filter water may not always be present. Low-cost, easily deployable alternatives that do not require sophisticated equipment and eliminate the need for time-consuming filtration, warrant investigation.

Here, we present an alternative eDNA collection approach that does not require active filtration.

By submerging secured membranes in the water column, we demonstrate the viability of passive eDNA collection. We hypothesized that both positively charged nylon and non-charged cellulose ester membranes can collect eDNA when placed in the water column through electrostatic attraction or entrapment, respectively. We use passive eDNA collection to characterize the fish diversity at both a temperate and tropical marine site,

comparing our results to those achieved through active filtration. We suggest that this inexpensive and more scalable method of eDNA sampling can increase biological replication in the field, permitting new types of ecological questions to be addressed.

## Results

**Passive eDNA collection enabled species detection**. To assess the viability of passively collecting eDNA, we submerged membranes below the ocean surface at both a tropical (Ashmore Reef) and temperate (Daw Island) site for comparison with eDNA collection via active filtration (Fig. 1). Fish taxa were detected from 141 of 146 passively deployed membranes (97%; 66/70 at Ashmore Reef and 77/78 at Daw Island) and on all 18 actively filtered membranes (100%; 9/9 at Ashmore Reef and 9/9 at Daw Island).

**Membrane material influenced species richness in tropical waters**. Two different membranes (charged nylon versus non-charged cellulose ester) were trialled to determine if the membrane material used during passive eDNA collection influenced fish detection. We detected significantly more fish species per non-charged membrane at Ashmore Reef (Kruskal–Wallis, $X^2 = 34.81$, d.f. = 1, $p < 0.001$; Fig. 2a, left), where the mean number of taxa was more than three times that detected on charged membranes (10 versus 3, respectively). In contrast, at Daw Island, there was no significant difference in species detection between membrane materials ($X^2 = 2.21$, d.f. = 1, $p < 0.14$; Fig. 2b, right). The mean number of fish detected per charged versus non-charged membrane was 8 and 11, respectively. Of the five membranes that yielded no fish eDNA, all were positively charged nylon (four from Ashmore Reef and one from Daw Island). For comparison, the mean number of fish taxa detected per non-charged actively filtered membrane (1 L of water) was 42 and 17 at Ashmore Reef and Daw Island, respectively.

Significant differences in initial copy number of DNA existed between all treatments at Ashmore Reef ($F = 12.29$, d.f. = 2, $p < 0.001$; Supplementary Fig. S1a, left), with actively filtered non-charged membranes containing the most DNA, and passively collected charged membranes containing the least (normality passed, Shapiro–Wilk, $W = 0.98$, $p = 0.31$). In contrast, at Daw Island, no significant differences in initial copy number of DNA existed between treatments ($X^2 = 1.91$, d.f. = 2, $p = 0.38$; Supplementary Fig. S1a, right).

**Increased submersion time did not increase species richness**. To examine whether increased submersion time of membranes led to increased species richness, we retrieved membranes after various length deployments (Fig. 1b). At Ashmore Reef, we found some evidence that submersion time affected fish detection (Table 1; $X^2 = 9.37$, d.f. = 3, $p = 0.02$, charged; $X^2 = 7.48$, d.f. = 3, $p = 0.06$, non-charged), where a significant difference was detected between charged filters deployed for 4 versus 8 h (mean species detection = 2 and 5, respectively; Dunn test with Bonferroni correction, $p = 0.014$). The initial copy number of DNA was higher on non-charged versus charged filter membranes collected after 8 h only (Wilcoxon test, $W = 67$, $p = 0.02$; Supplementary Fig. S1b, left). At Daw Island, we found no evidence that submersion time influenced the number of fish species detected (Table 2), regardless of membrane material ($X^2 = 6.14$, d.f. = 4, $p = 0.19$, charged; $X^2 = 1.96$, d.f. = 4, $p = 0.74$, non-charged), nor were any differences in initial copy number of DNA detected (Supplementary Fig. S1b, right).

**Passive eDNA collection characterized community composition**. To evaluate the effectiveness of passive eDNA collection, we

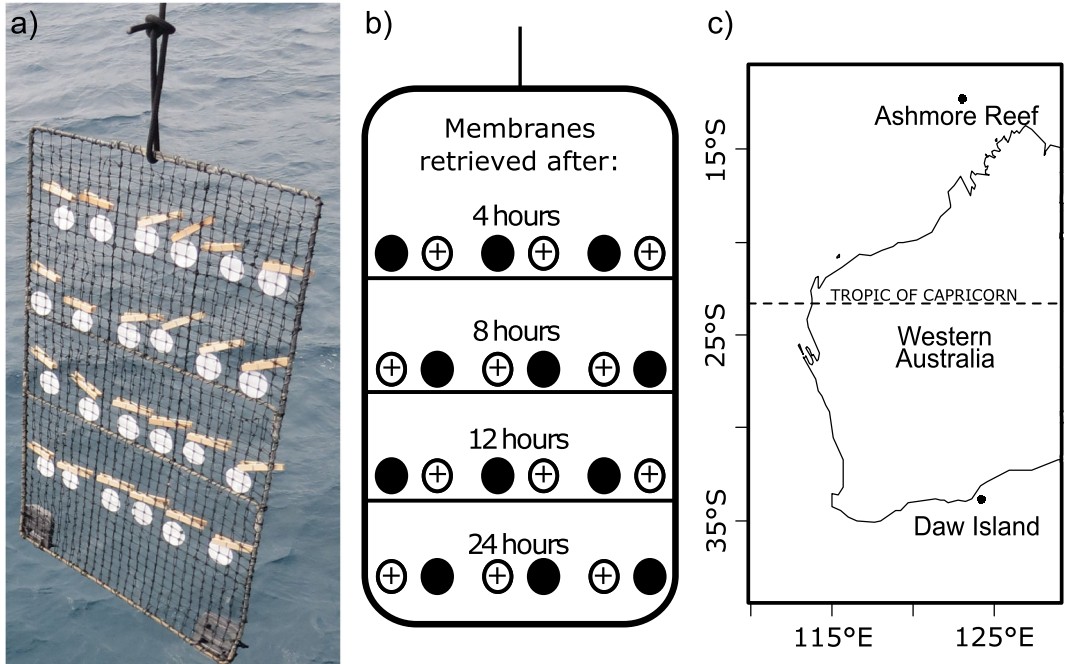

**Fig. 1 Passive eDNA collection experimental apparatus, design and location. a** Experimental apparatus containing filter membranes that were submerged 1 m below the ocean surface from the bow of a boat. **b** Experimental design showing non-charged cellulose ester (solid circles) and charged nylon (open circle with +) membrane deployment position and submersion duration. **c** Location of the deployment sites in the tropical (Ashmore Reef) and temperate (Daw Island) waters of Western Australia.

compared the species diversity obtained from passive eDNA collection to that achieved by the conventional method of actively filtering water samples. For all Ashmore Reef samples, we assigned 3,060,265 sequence reads to 172 fish taxa from 44 families (Supplementary Data 1). A total of 109 fish taxa were detected by passive sampling (45 on charged nylon and 100 on non-charged cellulose membranes), and 137 were detected using active filtration (Fig. 2a, left). Species composition was significantly different between all treatments (Fig. 2b, c for statistics; all $p$ values < 0.01), although the most abundant fish inhabiting Ashmore Reef[27], such as *Acanthurus triostegus* and *Halichoeres trimaculatus* (Table 1), were detected by all collection methods.

For all Daw Island samples, we assigned 5,259,198 sequence reads to 71 fish taxa from 39 families (Supplementary Data 2). A total of 68 fish taxa were detected by passive sampling (59 on charged nylon and 59 on non-charged cellulose membranes; Fig. 2a, right), and 53 were detected using active filtration. Species composition was similar for active filtration and non-charged passive collection (Fig. 2b), but significantly different between charged passive and active filtration (Fig. 2c for statistics; $p$ value > 0.01). A notable unique detection from a positively charged passive membrane was the apex predator *Charcharodon carcharias* (Table 2).

**Collection design for active filtration can influence variance in detection.** We found actively filtered samples displayed less variation in fish community at Ashmore Reef than Daw Island (Fig. 2b, grey dashed lines). We investigated this further and found that samples showed similarity by the time (Supplementary Fig. S2) and day (Supplementary Fig. S3) they were collected.

## Discussion
We developed and evaluated an alternative approach for the collection of aquatic eDNA that does not require active filtration. Our results provide compelling evidence that eDNA can be passively collected from marine waters with minimal equipment and without using granular materials[28], which require additional handling in the laboratory. This was true for two alternative membrane materials (positively charged nylon and non-charged cellulose ester) and for the detection of fish taxa in both tropical and temperate environments.

The promise of eDNA as a universal biomonitoring tool[29] has been realized in many respects for species detection and biodiversity studies, with the publication rate growing rapidly[6,9]. So far, however, the use of eDNA for the estimation of abundance and beta diversity has been more limited[30,31]. It is possible to use the number of samples in which an individual species is detected as a proxy measure of relative abundance, provided there is a high degree of biological replication[32]. Yet, this level of replication is rarely achieved with conventional active eDNA sampling and filtration[33]. Our passive eDNA collection method will allow for increased biological replication in the field, thereby permitting new types of ecological questions to be addressed[34]. Passive eDNA sampling will improve beta diversity estimates by increasing the number of biological replicates taken per area. Likewise, relative abundance, prevalence or biomass estimation through the frequency of occurrence metrics will be improved by collection of more samples.

Suspending membranes in the water column enabled eDNA collection and fish detection. To our knowledge, this is the first successful trial of passive eDNA collection directly from the ocean. Active filtration is the most common aquatic eDNA collection method used, in part, because it can process large water volumes required to detect rare species[11]. Yet, our results indicate that passive eDNA collection is a viable option for fish species detection, and in temperate ocean conditions, may even detect the same or greater overall species richness than achieved by active filtration. Although fewer fish taxa were detected on our submerged membranes at the tropical Ashmore reef site compared to active filtration (Fig. 2a), the richness of taxa detected by these methods was similar at the temperate site. Differences in taxa

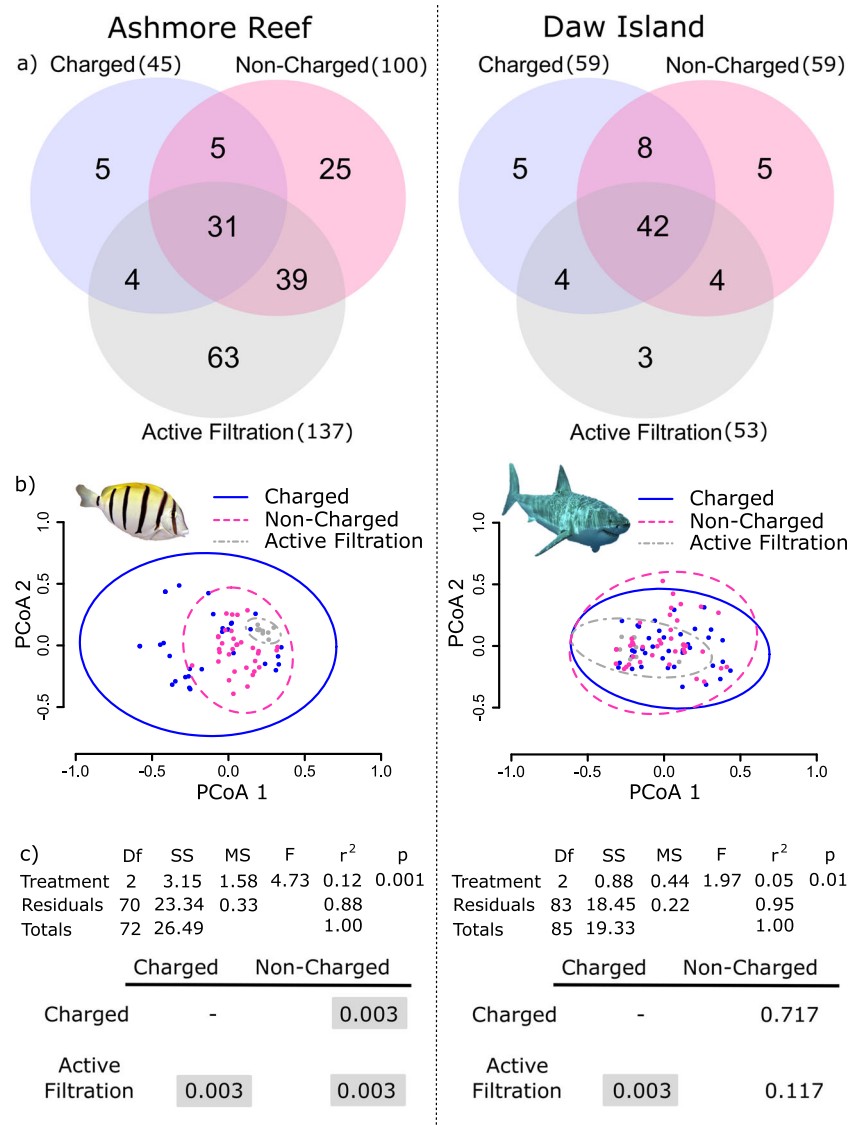

**Fig. 2 Detection of fish taxa by eDNA collection method. a** Venn diagrams representing the number of fish taxa detected at Ashmore Reef and Daw Island by passive eDNA collection methods with charged nylon (blue) and non-charged cellulose (pink) filter membranes, compared to actively filtering 9 L of water at each site (grey). **b** A principal coordinate analysis plot with 99% ellipses by eDNA capture method, where each dot represents a sample and colours correspond to eDNA capture method. **c** Pairwise comparisons by eDNA capture method (Bonferroni corrected; grey boxes indicate significance at $\alpha = 0.05$). Fish images obtained from Wikimedia Commons.

detection may reflect the physical limits of the membranes capacity to absorb eDNA passively, which may have been exceeded in the more species-rich tropical site, or not reached in the temperate waters. Such issues could be addressed through greater replication, just as increasing volume of filtered water increases species diversity identified by active collection[17], or through the use of alternative materials like activated carbon or clay[28].

Both membrane materials (charged nylon and non-charged cellulose ester) enabled passive eDNA collection, regardless of submersion time. Nevertheless, our charged nylon membranes failed to detect fish on five occasions, whereas all non-charged cellulose membranes detected fish. This may suggest that some materials outperform others for passive eDNA collection, and that the collection of eDNA on passive membranes may not be

dominated by electrostatic attraction of naked DNA molecules to the membrane, but by other mechanisms. Various membrane materials show different binding affinities for eDNA fragments[35]. For example, cellulose nitrate membranes show higher DNA yield than polycarbonate, polyethene sulfone and polyvinylidene fluoride, while glass microfiber filters outperform polycarbonate membranes[35–38]. More recently, in freshwater microcosm and field experiments, granular activated carbon was shown to capture an order of magnitude more eDNA than montmorillonite clay[28]. Trialling additional materials, either those with high binding affinities or dense surface areas, could help identify ways to increase DNA capture rate and improve detection efficiency, and would be aided by a mechanistic understanding of passive eDNA capture. It is likely that much of what we term 'eDNA' is DNA bound with other cellular components, so the properties of

**Table 1 Taxa detected at Ashmore Reef.**

| Family name | Taxon name | Passive filtration | | | | | | | | Active filtration |
|---|---|---|---|---|---|---|---|---|---|---|
| | | Charged | | | | Non-charged | | | | |
| | | 4 h | 8 h | 12 h | 24 h | 4 h | 8 h | 12 h | 24 h | |
| Acanthuridae | Acanthurus blochii | | | | | | | B | B | B |
| | Acanthurus lineatus | | | | | | | | | A |
| | Acanthurus triostegus | | B | B | B | B | B | B | B | B |
| | Acanthurus xanthopterus | | B | B | B | B | B | | B | B |
| | Ctenochaetus binotatus | | B | | B | | B | B | | B |
| | Ctenochaetus striatus | | B | | B | | B | B | | B |
| | Naso annulatus | | | | | | | | | A |
| | Naso brachycentron | | | | | | | P | | |
| | Naso unicornis | | | | | | | | | A |
| | Zebrasoma sp. | | | | | | | | | A |
| | Zebrasoma scopas | | | | | | | | | A |
| Albulidae | Albula glossodonta | | | | | B | B | B | | B |
| Apogonidae | Apogon sp. | | | | | | | B | B | B |
| | Jaydia sp. | | | | | | | B | | B |
| Arripidae | Arripis georgianus | | | | | P | | | | |
| Atherinidae | Hypoatherina temminckii | | | | B | B | B | | B | B |
| Balistidae | Balistapus undulatus | | | | | P | | P | P | |
| | Balistidae—unknown 1 | | | | | B | B | B | B | B |
| | Melichthys niger | | | | | | B | | | B |
| | Rhinecanthus aculeatus | | B | | B | B | B | B | B | B |
| | Rhinecanthus rectangulus | | | | | | | | | A |
| | Rhinecanthus verrucosus | | | | | | | | P | |
| Belonidae | Platybelone argalus | | | | | | | | | A |
| | Strongylura incisa | | | B | | | B | B | | B |
| | Tylosurus crocodilus | | B | | | B | B | B | B | B |
| Blenniidae | Aspidontus taeniatus | | | | | | | | | A |
| | Atrosalarias holomelas | | | | | | | | | A |
| | Blenniella periophthalmus | | | | B | B | | | B | B |
| | Cirripectes sp. | | | | | | | P | | |
| | Salarias fasciatus | | | | | | | B | B | B |
| Carangidae | Caranx ignobilis | | | B | B | B | | | | B |
| | Carangidae—unknown 1 | | B | | | | | B | B | B |
| Chaetodontidae | Chaetodon auriga | | | | | P | | | | |
| | Chaetodon vagabundus | | | | | | B | | | B |
| | Chaetodon sp. | | | | | | | | | A |
| Cirrhitidae | Paracirrhites forsteri | | | | | | | | | A |
| Clupeidae | Amblygaster sirm | | | | | | B | | | B |
| | Spratelloides delicatulus | B | B | B | B | B | B | B | B | B |
| | Spratelloides gracilis | | | | | | | P | | |
| Congridae | Gnathophis sp. | | | | | | | | | A |
| Ephippidae | Platax orbicularis | P | | | | P | | | | |
| | Platax teira | | B | B | B | | B | B | B | B |
| Fistulariidae | Fistularia commersonii | | | | | | | | | A |
| Gadidae | Micromesistius sp. | | | P | | | | | | |
| Gobiidae | Bryaninops sp. | | | | | | | | | A |
| | Eviota sp. 1 | | | | | | | | | A |
| | Eviota sp. 2 | | | | | B | B | B | | B |
| | Exyrias sp. | | | | | | | | | A |
| | Gobiodon sp. | | | | | | | | | A |
| | Gobiidae—unknown 1 | | | | | | | | | A |
| | Gobiidae—unknown 2 | | | | | | | | | A |
| | Gobiidae—unknown 3 | | | | | | P | | | |
| | Gobiidae—unknown 4 | | | | | | | | | A |
| | Paragobiodon sp. | | | | | | | | | A |
| | Vanderhorstia ornatissima | | | | | | | | | A |
| Haemulidae | Plectorhinchus chaetodonoides | | | | | B | B | B | B | B |
| Hemiramphidae | Hyporhamphus sp. | | | | | | | | B | B |
| Holocentridae | Myripristis botche | B | | | | B | | | B | B |
| | Myripristis murdjan | | | | | | | | | A |
| | Sargocentron rubrum | | | | | | | | | A |
| Kuhliidae | Kuhlia sp. | | | | | | | | | A |
| Kyphosidae | Kyphosus sp. | | | | | B | B | B | B | B |

**Table 1 (continued)**

| Family name | Taxon name | Passive filtration | | | | | | | | Active filtration |
|---|---|---|---|---|---|---|---|---|---|---|
| | | Charged | | | | Non-charged | | | | |
| | | 4 h | 8 h | 12 h | 24 h | 4 h | 8 h | 12 h | 24 h | |
| Labridae | *Anampses* sp. | | | | | | | | | A |
| | *Chlorurus* sp. | | | | | | | | | A |
| | *Choerodon schoenleinii* | | | | | | | | | A |
| | *Cirrhilabrus exquisitus* | | | | | | | | | A |
| | *Cymolutes praetextatus* | B | B | | | B | B | B | B | B |
| | *Epibulus* sp. | | P | | | | | | | |
| | *Gomphosus varius* | | | | | | P | | | |
| | *Halichoeres hortulanus* | | | | | | | | | A |
| | *Halichoeres melanurus* | | | | | | | | | A |
| | *Halichoeres nebulosus* | | | | | | | | | A |
| | *Halichoeres trimaculatus* | | B | B | | B | B | B | B | B |
| | *Hipposcarus longiceps* | | B | | | | B | B | | B |
| | *Labrichthys unilineatus* | | B | B | | B | | | | B |
| | *Novaculichthys taeniourus* | | | B | | | | | B | B |
| | *Pseudocheilinus evanidus* | | | | | | | P | | |
| | *Pseudocheilinus hexataenia* | | | | | | | B | | B |
| | *Scarus dimidiatus* | | | | | | | | | A |
| | *Scarus niger* | | | | | | | | | A |
| | *Scarus psittacus* | | | | | | | | | A |
| | *Scarus* sp. | | | | | | | | | A |
| | *Stethojulis bandanensis* | | | | | | B | B | | B |
| | *Stethojulis strigiventer* | | | | | | | B | | B |
| | *Stethojulis* sp. | | | | | | B | B | | B |
| | *Stethojulis trilineata* | | | | | P | | | | |
| | *Thalassoma amblycephalum* | | | | | B | | | | B |
| | *Thalassoma lunare* | | | | | | B | | B | B |
| | *Thalassoma* sp. | | | | | | B | | B | B |
| Lethrinidae | *Lethrinus nebulosus* | | | | | | | | | A |
| | *Lethrinus obsoletus* | | | | | | | | | A |
| | *Lethrinus* sp.1 | | | | | B | B | | B | B |
| | *Lethrinus* sp.2 | | | | | | | | | A |
| | *Lethrinus variegatus* | | | | | | | | | A |
| | *Monotaxis grandoculis* | | | | | | | | | A |
| Lutjanidae | *Aprion virescens* | B | | | | B | B | B | | B |
| | *Caesio caerulaurea* | | | | | | | P | | |
| | *Caesio xanthonotus* | | | | | | | | | A |
| | *Lutjanus bohar* | | B | B | | B | B | B | B | B |
| | *Lutjanus decussatus* | | | | | | | | | A |
| | *Lutjanus fulvus* | | | | | | | P | | |
| | *Lutjanus kasmira* | | | | | | | | B | B |
| | *Lutjanus* sp. 1 | | B | B | | B | B | B | B | B |
| | *Lutjanus* sp. 2 | B | | | | B | B | B | B | B |
| | *Pterocaesio* sp. | B | B | | | B | | | | B |
| | *Pterocaesio tile* | | | | | | | | | A |
| Monacanthidae | *Cantherhines dumerilii* | | | | | P | P | | | |
| | *Monacanthus chinensis* | P | | | | | | | | |
| Mugilidae | *Crenimugil crenilabis* | | | | | | P | | | |
| | *Ellochelon vaigiensis* | | P | | | | | P | | |
| | *Mulloidichthys vanicolensis* | | | | | B | B | | B | B |
| | *Parupeneus barberinus* | | | | | | | | | A |
| | *Parupeneus chrysopleuron* | | | | | P | | | | |
| | *Parupeneus multifasciatus* | | | | | | | | | A |
| | *Parupeneus* sp. | B | B | | | | B | | B | B |
| | *Upeneus tragula* | P | | | | | | | | |
| Muraenidae | *Echidna nebulosa* | | | | | | B | B | | B |
| | *Gymnothorax buroensis* | | | | | | | | | A |
| | *Gymnothorax flavimarginatus* | | | | | | | | | A |
| | *Gymnothorax* sp. | | | | | B | B | B | | B |
| Myctophidae | *Diaphus watasei* | | | | | | | | | A |
| | Myctophidae—unknown 1 | | | | | | | | | A |
| Myliobatidae | *Aetobatus ocellatus* | B | B | B | B | B | B | B | B | B |
| Platycephalidae | *Sunagocia otaitensis* | | | | | | P | | | |
| | *Plesiops* sp. | | | | | | B | B | | B |

**Table 1 (continued)**

| Family name | Taxon name | Charged | | | | Non-charged | | | | Active filtration |
|---|---|---|---|---|---|---|---|---|---|---|
| | | 4 h | 8 h | 12 h | 24 h | 4 h | 8 h | 12 h | 24 h | |
| Pomacentridae | *Abudefduf sexfasciatus/vaigiensis* | | | | | B | B | B | | B |
| | *Acanthochromis* sp. | | | | | | | | | A |
| | *Amblyglyphidodon curacao* | | | | | | | | | A |
| | *Chromis atripectoralis* | | B | | | | | | | B |
| | *Chromis atripes* | | | | | | | | | B |
| | *Chromis lepidolepis* | | | | | | P | | | |
| | *Chromis ternatensis* | | B | | | | | | | B |
| | *Chromis* sp. | | | | | B | | B | | B |
| | *Chromis viridis* | | | | B | | | | | B |
| | *Chrysiptera glauca* | | B | | B | B | | | | B |
| | *Chrysiptera rex* | | | | | | | | | A |
| | *Chrysiptera* sp. | | B | | | | B | B | B | B |
| | *Dascyllus aruanus* | | | | | | | B | | B |
| | *Dascyllus reticulatus* | | | | | P | P | | | |
| | *Dascyllus trimaculatus* | | | P | | | P | | | |
| | *Dischistodus prosopotaenia* | | | | | | | | | A |
| | *Hemiglyphidodon plagiometopon* | | | | | | | | | A |
| | *Neopomacentrus* sp. | | | | | | | | | A |
| | *Plectroglyphidodon lacrymatus* | | | | | | | | | A |
| | *Plectroglyphidodon leucozonus* | | | | | B | | | | B |
| | *Pomacentrus bankanensis* | | | | | | | | P | |
| | *Pomacentrus lepidogenys* | | | | | | | B | B | B |
| | *Pomacentrus pavo* | | | | | B | | B | B | B |
| | *Pomacentrus* sp.1 | | | | B | | | B | B | B |
| | *Pomacentrus* sp. 2 | | | | | | | | | A |
| | *Stegastes fasciolatus* | | | | | | | | | A |
| | *Stegastes nigricans* | | | | B | | | B | | B |
| Pseudochromidae | *Pseudochromis* sp. | | B | | | B | | | | B |
| Rhynchobatidae | *Rhynchobatus* sp. | | P | | | P | P | P | P | |
| Schindleriidae | *Schindleria* sp. | | | | | | | | P | |
| Scombridae | *Auxis* sp. | | | | | | | P | | |
| | *Euthynnus* sp. | | | | | | | | | A |
| | *Scomberomorus commerson* | | | | | | | | | A |
| Scorpaenidae | *Scorpaenodes guamensis* | | | | | | | | P | |
| Serranidae | *Aethaloperca rogaa* | | | | | | | B | | B |
| | *Cephalopholis argus* | | | | | B | | B | | B |
| | *Cephalopholis leopardus* | | | | | | P | | | |
| | *Epinephelus* sp. | | | | | | B | | | B |
| | *Plectropomus laevis* | B | B | | B | | | B | B | B |
| | *Pseudogramma polyacanthus* | | | | | | | | | A |
| | *Variola louti* | | | | | | B | B | B | B |
| Soleidae | *Pardachirus pavoninus* | | | | | | | P | | |
| Synodontidae | *Trachinocephalus myops* | | | | | | | | P | |
| Tetraodontidae | *Arothron mappa* | | | | | | | | B | B |
| | *Arothron stellatus* | | | | | | | | B | B |
| Xiphiidae | *Xiphias gladius* | | P | | | | | | P | |

Taxa at Ashmore Reef detected by passive filtration treatment and submersion time (h) compared to those detected by active filtration. An A indicates taxa detected by active filtration only, P denotes those detected by passive filtration only and B indicates taxa detected by both methods.

pure DNA (e.g. negatively charged backbone) may be less important in determining eDNA recovery rates than expected.

Our passive eDNA collection method eliminates the need to collect and filter water. The two main benefits of eliminating the filtration step is reduced time spent sampling and no requirement for expensive equipment or power. Time spent filtering water could be used to deploy multiple passive membranes, which in turn would enable large-scale sampling and increased replication. For example, in the same time it took us to collect and filter a 1 L water sample at our anchor site aboard the vessel, we could deploy and retrieve all 24 membranes using our pearl frame apparatus. Although the volume of water passing over the membrane is unknown, flow metres could be used to provide a proxy if necessary for the question being addressed. Passive eDNA collection does, however, require anchoring and ensuring membranes are secured in place. We used a pearl oyster aquaculture frame with mesh pockets, which illustrates how low-cost options for secure deployment can be easily imagined.

The most compelling value of passive aquatic eDNA collection is that it enables a broader range of ecological questions to be addressed by increasing biological sampling. A major shortcoming of most aquatic eDNA surveys is their low level of biological replication[16,32]. The practical reasons for this are the time and cost constraint imposed by actively filtering water. Consequently, eDNA metabarcoding studies may be effective for some biodiversity assessments but are generally under-sampled for beta

**Table 2 Taxa detected at Daw Island.**

| Family name | Taxon name | Passive filtration | | | | | | | | | | Active filtration |
| --- | --- | --- | --- | --- | --- | --- | --- | --- | --- | --- | --- | --- |
| | | Charged | | | | | Non-charged | | | | | |
| | | 4 h | 8 h | 12 h | 24 h | 34 h | 4 h | 8 h | 12 h | 24 h | 34 h | |
| Aplodactylidae | Aplodactylus sp. | B | | B | B | B | B | B | B | B | B | B |
| Arripidae | Arripis georgianus | B | B | B | B | B | B | B | B | B | B | B |
| | Arripis sp. | B | B | B | | | B | B | B | B | | B |
| Aulopidae | Latropiscis purpurissatus | | B | | B | | B | | | | | B |
| Berycidae | Centroberyx sp. | P | | | | | | | | | | |
| Bothidae | Lophonectes sp. | | | | | | | | P | | | |
| Callionymidae | Repomucenus calcaratus | P | | | | | P | | P | P | | |
| Carangidae | Pseudocaranx sp. | B | B | B | B | | B | B | B | B | | B |
| | Pseudocaranx wrighti | | B | | | | | | | | | B |
| | Seriola lalandi | B | B | B | | B | B | B | B | | B | B |
| | Trachurus sp. | | | | B | | | | | | | B |
| Cheilodactylidae | Cheilodactylus sp. | P | P | | | | P | | | | | |
| | Cheilodactylidae—unknown 1 | | | | | | P | P | | P | | |
| | Nemadactylus valenciennesi | | | | | | | | | P | | |
| Chironemidae | Chironemus georgianus | P | | P | | | P | P | | | P | |
| | Chironemus maculosus | | B | B | B | | | | B | B | | B |
| Clinidae | Heteroclinus adelaidae | B | | | B | | | | B | B | | B |
| | Heteroclinus eckloniae | | | | | B | B | | | | | B |
| Clupeidae | Clupeidae—unknown 1 | B | B | | | | | B | | | | B |
| | Sardinops sagax | B | B | B | B | B | B | B | B | B | B | B |
| Congridae | Gnathophis longicauda | | | | B | | | | | | | |
| Dasyatidae | Bathytoshia brevicaudata | | B | B | B | | | | | | | B |
| Dinolestidae | Dinolestes lewini | | | | B | | B | | | | | A |
| Dussumieriidae | Etrumeus jacksoniensis | B | | B | B | | | B | | B | | B |
| Engraulidae | Engraulis australis | B | | | | | | B | B | B | | B |
| Enoplosidae | Enoplosus armatus | | B | B | | | | | | B | | B |
| Gerreidae | Parequula melbournensis | | | | | | | B | | B | | B |
| Hemiramphidae | Hyporhamphus melanochir | B | | B | | B | B | B | B | B | B | B |
| Isonidae | Iso rhothophilus | | | | | | | | | | | A |

**Table 2 (continued)**

| Family name | Taxon name | Passive filtration | | | | | | | | | | Active filtration |
|---|---|---|---|---|---|---|---|---|---|---|---|---|
| | | Charged | | | | | Non-charged | | | | | |
| | | 4 h | 8 h | 12 h | 24 h | 34 h | 4 h | 8 h | 12 h | 24 h | 34 h | |
| Kyphosidae | Girella sp. | B | B | B | B | | B | B | B | B | B | B |
| | Kyphosus gladius/ sydneyanus | B | B | B | B | | B | B | B | B | B | B |
| | Scorpis sp. | B | B | | B | B | B | B | B | B | | B |
| Labridae | Achoerodus sp. | B | B | B | B | | B | B | B | | B | B |
| | Australabrus maculatus | B | B | B | | | B | B | B | B | | B |
| | Bodianus sp. | | | | P | | P | | | | | |
| | Eupetrichthys angustipes | B | B | B | P | | B | B | B | B | | B |
| | Halichoeres brownfieldi | B | B | B | B | | B | B | B | B | B | B |
| | Labridae— unknown 1 | | | B | B | | | | B | | | B |
| | Notolabrus fucicola | | B | B | | | B | | | | B | B |
| | Notolabrus parilus | | B | B | | | B | | B | B | B | B |
| | Ophthalmolepis lineolata | | B | B | B | | B | | B | | | B |
| | Pictilabrus laticlavius | B | B | B | B | | B | B | B | B | B | B |
| Lamnidae | Carcharodon carcharias | | P | | | | | | | | | |
| Monacanthidae | Acanthaluteres sp. | B | B | | | | | B | B | B | B | B |
| | Monacanthidae— unknown 1 | | | B | | | B | | B | | B | B |
| | Nelusetta ayraudi | P | | P | | | | | P | | | |
| | Scobinichthys granulatus | | | | | | | | B | | | B |
| Moridae | Lotella rhacina | B | | | | | | | | | | B |
| | Pseudophycis barbata | | | | | | | | | | | A |
| Mullidae | Upeneichthys sp. | B | B | | B | | B | B | B | B | B | B |
| | Upeneichthys stotti | B | | | B | B | | B | B | B | | B |
| Myliobatidae | Myliobatis australis | | | B | | | B | | B | | | B |
| Odacidae | Heteroscarus acroptilus | B | | B | B | | | | B | B | | B |
| | Olisthops cyanomelas | B | B | B | B | | B | B | B | B | B | B |
| Pempheridae | Siphonognathus sp. | B | B | | B | | B | B | B | B | | B |
| | Parapriacanthus elongatus | B | B | B | B | | B | B | B | B | | B |
| Pinguipedidae | Pempheris sp. | B | B | B | B | | B | B | | | B | B |
| | Parapercis haackei | B | B | B | | | | B | | | | B |
| | Parapercis ramsayi | | P | P | | | | B | | | | A |

**Table 2 (continued)**

| Family name | Taxon name | Passive filtration — Charged 4 h | 8 h | 12 h | 24 h | 34 h | Non-charged 4 h | 8 h | 12 h | 24 h | 34 h | Active filtration |
|---|---|---|---|---|---|---|---|---|---|---|---|---|
| Platycephalidae | *Leviprora inops* | P | | | | | | | | | | |
| | *Platycephalus grandispinis* | B | | | B | | | B | B | B | | B |
| Pomacentridae | *Chromis* sp. | P | | | P | | P | | | | | B |
| | *Parma microlepis* | B | B | | | | B | B | B | B | P | B |
| Scombridae | Scombridae—unknown 1 | | | | | | B | | | | | |
| | *Scomber* sp. | B | B | | B | | B | B | | B | | B |
| Scorpaenidae | Scorpaenidae—unknown 1 | | | | | | P | | | | | |
| Serranidae | *Acanthistius* sp. | | | | P | | P | P | P | | | |
| Sphyraenidae | *Sphyraena novaehollandiae* | | | | P | | P | | P | | | |
| Tetraodontidae | *Omegophora armilla* | | B | B | | | | | B | | B | B |
| Triglidae | *Lepidotrigla* sp. | B | B | | B | | B | B | B | | B | B |
| Xiphiidae | *Xiphias gladius* | | | | | | | P | P | | | |

Taxa at Daw Island detected by passive filtration treatment and submersion time (h) compared to those detected by active filtration. An A indicates taxa detected by active filtration only, P denotes those detected by passive filtration only and B indicates taxa detected by both methods.

diversity measurement[39]. The disconnect between sampled eDNA and the live organisms that produced it make abundance estimates by eDNA particularly challenging[40]. A more widely accepted approach is to treat detection of any amount of a taxon's DNA as a 'presence' and to collect enough biological samples to enable comparative frequency analysis between treatments[1,29,41]. This is exactly the type of analysis that is constrained by the inefficiencies of active water eDNA sampling by filtration. Passive eDNA collection provides an immediate solution to the problem by enabling increased replication and large-scale sampling. Frequency-based semi-quantification is inherently constrained by sample size[42]. The ability to collect large numbers of samples (thereby increasing sample size) from each site makes analysis by frequency of occurrence of a DNA sequence between sites powerful. Collection of many samples from microhabitats within each site also allows for fine-scale distribution mapping of species where presence is inferred by a DNA sequence. The far greater sampling rate allowed by passive eDNA sampling would also enable a greater temporal sampling density so that the extent of residence of species might be inferred in systems with rapidly mixing water, or fast degradation of eDNA signal[43]. Finally, gamma biodiversity estimation will be improved through passive eDNA sampling of a range of microhabitats within a field site because each microenvironment produces a different alpha biodiversity[9]. Comparisons of biodiversity in space and time and in relation to both natural and anthropogenic environmental variation is at the core of biodiversity monitoring and these more in-depth characterizations must underpin the next generation of biological monitoring[29].

## Methods

**Study site.** Sampling was conducted in the tropical waters of Ashmore Reef (122° 58.99′ E, 12°14.27′S) and the temperate waters surrounding Daw Island (124°07.86′ E, 33°51.01′ S, Fig. 1c). Ashmore Reef lies 320 km off the north-west coast of Western Australia and is 25 km long and 12 km wide, comprising three low lying uninhabited islands, two lagoons, extensive reef and sand flat habitats and seagrass meadows. Ashmore Reef was established as an Australian marine park in 1983. Sampling took place in surface waters at the vessel mooring site, where the depth was <10 m. Daw Island lies 35 km off the south coast of Western Australia and is part of the Eastern Recherche Marine Park, which was established in 2012. Macroalgae habitats surround the island, and the temperate waters of this area are known for their high species endemicity. Sampling at Daw Island took place in the surface waters at the vessel mooring site, where the depth was <20 m.

**Limiting contamination.** Contamination was minimized by wearing gloves and using sterile tweezers to handle each filter membrane. All membrane filters were immediately frozen after collection and stored at −20 °C until further processing in the laboratory. To further account for possible contamination in the field during active filtration at Daw Island, we filtered 500 mL of deionized water at the start of every day. Prior to use, all collection and deployment apparatus were sterilized by soaking in 10% bleach solution for at least 15 min and rinsed in deionized water.

**Passive eDNA collection.** We trialled passive eDNA collection by submerging two membrane materials ~1 m below the ocean surface in the mesh pockets of a pearl oyster aquaculture frame and collected them at specified intervals (Fig. 1a). One membrane material was a positively charged electrostatic nylon (0.45 μm Biodyne™ B, 47 mm), while the other was a non-charged, cellulose ester (0.45 μm Pall GN-6 Metricel®). We used a positively charged membrane because extracellular DNA is negatively charged due to the phosphate on the sugar phosphate backbone. All membranes were certified sterile upon purchase.

To examine whether increased submersion time of membranes led to increased detection, we retrieved triplicate membranes after 4, 8, 12 and 24 h of deployment (Fig. 1b). This design was deployed at both our Ashmore Reef and Daw Island sites (Fig. 1c). Membranes were deployed over a 3-day period (17–20 June 2019) at Ashmore Reef (3 days × 24 membranes − 4 membranes were lost during retrieval due to handling error; *n* = 68 membranes) and 5 days (29 January to 2 February 2019) at Daw Island (3 days × 24 membranes + 6 membranes deployed for 34 h; *n* = 78 membranes). Time allowed for the addition of a 34-h deployment trial at Daw Island.

**Active eDNA collection.** We collected water for active eDNA filtration to compare to our passive method. Nine 1 L surface water samples were collected in sterile 1 L

containers at both sites and filtered over nine non-charged cellulose membranes (47 mm diameter, 0.45 μm pore size) using a peristaltic Sentino® Microbiology Pump in a laboratory setting aboard the vessel. At Ashmore reef, we actively filtered triplicate 1 L water samples at 8:00, 12:00 and 16:00 on the final day of the experiment (20 June 2019). At Daw Island, we actively filtered 1 L samples at 08:00, 12:00 and 16:00 each day for all 3 days of the experiment (30 January–1 February 2019).

**eDNA extraction from membranes**. Total nucleic acid was extracted from the whole membrane using a DNeasy Blood and Tissue Kit (Qiagen; Venlo, The Netherlands), with an additional 40 μL of proteinase K used during a 3-h digestion period at 56 °C on rotation (300 r.p.m.). DNA was eluted in 200 μL AE buffer. All extractions took place in a dedicated DNA extraction laboratory using a QIAcube (Qiagen; Venlo, The Netherlands), where benches and equipment were routinely bleached and cleaned.

**DNA metabarcode amplification from eDNA**. One-step quantitative polymerase chain reactions (qPCR) were performed in duplicate for each sample using 2 μL of extracted DNA and a mitochondrial DNA 16S ribosomal DNA universal primer set targeting fish taxa (16SF/D, 5′-GACCCTATGGAGCTTTAGAC-3′ and 16S2R-degenerate, 5′-CGCTGTTATCCCTADRGTAACT-3′)[44,45], with the addition of fusion tag primers unique to each sample that included Illumina P5 and P7 adaptors. We chose to analyse fish eDNA because this is one of the most common assessments made in aquatic eDNA metabarcoding studies[6,8–11] and is associated with a substantial reference database for taxonomic identification. A single round of qPCR was performed in a dedicated PCR laboratory. qPCR reagents included 5 μL AllTaq PCR Buffer (Qiagen; Venlo, The Netherlands), 0.5 μL AllTaq DNA polymerase, 0.5 μL dNTPs (10 mM), 1.0 μL Ultra BSA (500 μg/μL), SYBR Green I (10 U/μL), 0.5 μL forward primer (20 μM) and 5.0 μL reverse primer (20 μM), 2 μL of DNA and Ultrapure™Distilled Water (Life Technologies) made up to 25 μL total volume. Mastermix was dispensed manually and qPCR was performed on a CFX96 Touch™ Real-Time PCR Detection System (Bio-Rad, California, USA) using the following conditions: initial denaturation at 95 °C for 5 min, followed by 40 cycles of 30 s at 95 °C, 30 s at the primer annealing temperature 54°C and 45 s at 72 °C, with a final extension for 10 min at 72 °C. All duplicate qPCR products from the same subsample were combined prior to library pooling. The mean Cq value from qPCR duplicates was used as an indication of initial DNA copy number. Two sequencing libraries were made by pooling amplicons into equimolar ratios based on qPCR Ct values and sequenced on an Illumina Miseq platform (Illumina, San Diego, USA); one for Daw Island samples and the other for Ashmore Reef samples. The libraries were size selected using a Pippin Prep (Sage Science, Beverly, USA) and purified using the Qiaquick PCR Purification Kit (Qiagen, Venlo, The Netherlands). The volume of purified library added to the sequencing run was determined by quantifying the concentration[46] using a Qubit 4 fluorometer (Thermo Fisher Scientific). The library was unidirectionally sequenced using a 300 cycle MiSeq® V2 Reagent Kit and standard flow cell.

PCR plates included blank laboratory extraction controls (extraction reagents used with no DNA template), PCR-negative controls (2 μL of deionized water used rather than DNA template) and positive controls (dhufish; *Glaucosoma hebraicum*). Dhufish were an appropriate positive control for the tropical Ashmore samples since they are a subtropical fish species that do not occur at Ashmore reef. Dhufish do naturally occur in the vicinity of Daw Island at low densities, although no Dhufish were detected in any Daw Island samples other than positive controls. In addition, negative field controls were also conducted at Daw Island by filtering 500 mL of deionized water in situ. No negative control (extraction, PCR or field control) contained more than five reads for any fish species, except for one PCR-negative control which contained 11 reads. All positive controls amplified multiple reads identifying dhufish with 100% identity. However, 36 reads of the positive control showed up in two Ashmore Reef samples. Therefore, to ensure a conservative approach to detection efficiency, we required a minimum of 40 reads to count a fish species as present.

**DNA sequence data processing**. Data generated by Illumina sequencing were quality controlled using a series of steps prior to taxonomic assignment. First, the OBITools (https://pythonhosted.org/OBITools/) command 'ngsfilter' was used to assign each sequence record to the corresponding sample based on tag and primer. Then, 'obiuniq' was used to dereplicate reads into unique sequences. Reads <190 bp and with counts <10 were discarded. A denoising step was performed using 'obiclean' to retain only sequences with no variants containing a count >5% of their own. Each unique sequence was queried against the NCBI[47] nucleotide database using BLASTn[48]. The search set used in BLASTn was the nucleotide collection (nr/nt), with the programme selection optimized for highly similar sequences. Sequences were assigned to taxa using 'ecotag' and a result table was generated using 'obiannotate'. This sequence processing directly follows the procedure described at https://pythonhosted.org/OBITools/wolves.html. Our reference database built in silico using our universal fish primer assay (on 27 August 2020) is provided.

**Taxonomic assignment of DNA metabarcode amplicons**. The taxonomic assignment of BLASTn search hits for each sequence was resolved to species, genus and family based on the percent similarity (identities) to taxa alignments. A summary of assigned taxon name, identities, GenBank accession number and number of reads is provided for each unique sequence, ensuring complete transparency in taxonomic assignment (Supplementary Data 1 and 2). In general, taxonomic assignments were designated to species for identities >97%, to genus for identities <97% but >90% and to family for identities <90%. Sequences with <80% identity matches were not used. Taxon alignments were checked to ensure that the species assigned were known to the region (e.g. Atlas of Living Australia, http://www.ala.org.au; Fishes of Australia, http://fishesofaustralia.net.au/ and Fishbase, http://www.fishbase.org) and the Codes for Australian Aquatic Biota (CAAB number) are also reported.

**Statistics and reproducibility**. Standard parametric or non-parametric statistics were used to identify differences between treatments depending on normality (Shapiro–Wilks test, analysis of variance and Tukey's honest significant differences, Kruskal–Wallis test, Dunn test). Fish communities detected by each treatment were subjected to principal coordinate analysis using a Sorensen pair-wise dissimilarity matrix based on the presence/absence of taxa (APE and BETAPART)[49,50]. An analysis of variance on the dissimilarity matrix (adonis) was used to determine if treatment was a significant source of variation (VEGAN)[51] in the fish community composition. Pairwise comparisons (pairwise.perm.manova; RVAideMemoire)[52] were then used with Bonferroni adjustment to reveal which treatments were significantly different ($\alpha = 0.05$). Most statistics and graphics were produced using R (version 2.14.0; R Development Core Team 2011) and Inkscape (https://inkscape.org/), except for the Venn diagrams that were produced online (http://bioinformatics.psb.ugent.be/webtools/Venn/). Our methods contain all the necessary information required to repeat these experiments, including sample sizes.

**Reporting summary**. Further information on research design is available in the Nature Research Reporting Summary linked to this article.

## Data availability
A spatial representation of the eDNA survey results is available through the Atlas of Living Australia at https://collections.ala.org.au/public/show/dr16844.

Raw sequences, bioinformatic script, reference database and the final datasets are available on the CSIRO Data Access Portal at https://data.csiro.au/collections/collection/CIcsiro:46025v1.

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

## Acknowledgements

This project was funded by the CSIRO Environomics Future Science Platform. Ship time funding was provided by the Minderoo Foundation (Daw Island) and Parks Australia (Ashmore Reef). We thank the incredible crew and staff of the Pangaea Ocean Explorer (Flourishing Oceans, Minderoo Foundation) and the Kuri Pearl II (Bass Marine, Terrafirma Offshore). We also thank Charlette Brun for her assistance with sample preparation in the laboratory, Bruce Deagle for providing valuable comments on the manuscript and three anonymous reviewers for their constructive contributions that greatly improved the manuscript.

## Author contributions

C.B., S.N.J. and O.B.—designed study and/or provided intellectual direction; C.B., J.K., H. M. and T.St.—organized/participated in field trips and sample acquisition; C.B. and T.Si. —performed molecular research; C.B.—processed and analysed all data; C.B.—conducted the statistical analysis, produced graphics and tables; C.B., J.K., S.N.J. and O.B.—assisted with manuscript writing. All authors contributed to manuscript revisions

## Competing interests

The authors declare no competing interests.
