## [Peer Review File · Communications Biology]

Reviewers' comments:

Reviewer #1 (Remarks to the Author):

Overview

The authors presents a study that utilizes passive filter collection of environmental DNA and show that the results are comparable to active filtering methods. While the study is relevant to eDNA research, many of the statements regarding the differences between methodologies and the reason for using passive filtration could be improved and further supported throughout the manuscript in order to convince the reader that the methodology is a valid alternative to their current methods. Major considerations should be made to the structure of the introduction and discussion (comments below).

Abstract

Line 31: As compared to what?

Both passive and active filtration require water to pass through a membrane. There are pros and cons with either methods, such as losing the passive membrane prior to retrieval or potential contamination from placing the filter directly in the sampling environment. Observational wise I would expect similar results (provided that the methods allowed for sufficient material collection), but I would be more concerned with contamination via the passive filtration.

Introduction

Line 37 – 40 Thomese and willerslev proposed the orgin of eDNA. There have studies since that have assessed eDNA orgin and size that provide an updated view on eDNA state

Jo, T., Arimoto, M., Murakami, H., Masuda, R., & Minamoto, T. (2019). Particle Size Distribution of Environmental DNA from the Nuclei of Marine Fish. *Environmental Science & Technology*, 53(16), 9947–9956. <https://doi.org/10.1021/acs.est.9b02833>

Moushomi, R., Wilgar, G., Carvalho, G., Creer, S., & Seymour, M. (2019). Environmental DNA size sorting and degradation experiment indicates the state of *Daphnia magna* mitochondrial and nuclear eDNA is subcellular. *Scientific Reports*, 9(1). <https://doi.org/10.1038/s41598-019-48984-7>

Wilcox, T. M., McKelvey, K. S., Young, M. K., Lowe, W. H., & Schwartz, M. K. (2015). Environmental DNA particle size distribution from Brook Trout (*Salvelinus fontinalis*). *Conservation Genetics Resources*, 7(3), 639–641. <https://doi.org/10.1007/s12686-015-0465-z>

Line 40: Environmental DNA is not a technique, but rather encompasses a field of research

Deiner, K., Bik, H. M., Mächler, E., Seymour, M., Lacoursière-Roussel, A., Altermatt, F., Creer, S., Bista, I., Lodge, D. M., de Vere, N., Pfrender, M. E., & Bernatchez, L. (2017). Environmental DNA metabarcoding: Transforming how we survey animal and plant communities. *Molecular Ecology*. <https://doi.org/10.1111/mec.14350>

Bohmann, K., Evans, A., Gilbert, M. T. P., Carvalho, G. R., Creer, S., Knapp, M., Yu, D. W., & de Bruyn, M. (2014). Environmental DNA for wildlife biology and biodiversity monitoring. *Trends in Ecology & Evolution*, 29(6), 358–367. <https://doi.org/http://dx.doi.org/10.1016/j.tree.2014.04.003>

Line 41-42: This line needs support. See:

Seymour, M., Edwards, F. K., Cosby, B. J., Kelly, M. G., de Bruyn, M., Carvalho, G. R., & Creer, S. (2020). Executing multi-taxa eDNA ecological assessment via traditional metrics and interactive networks. *Science of The Total Environment*, 729, 138801.

<https://doi.org/10.1016/j.scitotenv.2020.138801>

Line 46-51: This section could be revised as several of the statements could be increased in accuracy. E.g., the use of a pumping mechanism is not universal for active filtration. For examples see:

Deiner, K., Walser, J.-C., Mächler, E., & Altermatt, F. (2015). Choice of capture and extraction methods affect detection of freshwater biodiversity from environmental DNA. *Biological Conservation*, 183, 53–63. <https://doi.org/http://dx.doi.org/10.1016/j.biocon.2014.11.018>

Deiner, K., Lopez, J., Bourne, S., Holman, L., Seymour, M., Grey, E. K., Lacoursière, A., Li, Y., Renshaw, M. A., Pfrender, M. E., Rius, M., Bernatchez, L., & Lodge, D. M. (2018). Optimising the detection of marine taxonomic richness using environmental DNA metabarcoding: the effects of filter material, pore size and extraction method. *Metabarcoding and Metagenomics*, 2. <https://doi.org/10.3897/mbmg.2.28963>

Line 53-57: These statements are rather vague and could use more support. See

Seymour, M., Durance, I., Cosby, B. J., Ransom-Jones, E., Deiner, K., Ormerod, S. J., Colbourne, J. K., Wilgar, G., Carvalho, G. R., de Bruyn, M., Edwards, F., Emmett, B. A., Bik, H. M., & Creer, S. (2018). Acidity promotes degradation of multi-species environmental DNA in lotic mesocosms. *Communications Biology*, 1(1). <https://doi.org/10.1038/s42003-017-0005-3>

Deiner, K., & Altermatt, F. (2014). Transport Distance of Invertebrate Environmental DNA in a Natural River. *PLoS ONE*, 9(2), e88786. <http://dx.doi.org/10.1371/journal.pone.0088786>

Line 58: Most studies do not filter 20L and studies have also shown that 1-2L is sufficient.

Mächler, E., Deiner, K., Spahn, F., & Altermatt, F. (2016). Fishing in the Water: Effect of Sampled Water Volume on Environmental DNA-Based Detection of Macroinvertebrates. *Environmental Science & Technology*, 50(1), 305–312. <https://doi.org/10.1021/acs.est.5b04188>

Aims and objective should be clarified and hypotheses better supported via the introduction

Methods

Lines 231-241: A lot of information is missing or is out of order. How many samples? Replications? Filter sizes? Length of passive filtration (particularly since you argue that the method saves time)? Active filtration equipment used? Were the samples preserved in any way or were they dried? What material was the sampling equipment? How was the equipment sterilized prior to each sampling event?

Line 253: Did you manage to get sufficient DNA concentrations with 200 μ L elutions? Many studies use lower elution volumes given the low concentration of eDNA in their extracts

Results

Line 78-80: How many samples and what was the interval. Don't make the reader go digging through figures to guess what the study design was.

Line 84-86: Consider an alternative statement than choosing to analyze fish because they are popular

Line 103-104: More details pertaining to the sampling apparatus construction and usage should be provided. How were the filters kept in place while they collected material? Was there any consideration for tidal or current influences?

Line 121: This pump is a benchtop model. Were the samples transported to a lab prior to filtering?

Line 132-134: This belongs in the discussion

Line 144-151: This belongs in the discussion as currently written

Discussion

Line 158-161: This is not supported presently, see

Iliana, B., R., C. G., Min, T., Kerry, W., Xin, Z., Mehrdad, H., Shadi, S., Mathew, S., David, B., Shanlin, L., Martin, C., & Simon, C. (2018). Performance of amplicon and shotgun sequencing for accurate biomass estimation in invertebrate community samples. *Molecular Ecology Resources*, 0(0). <https://doi.org/10.1111/1755-0998.12888>

Yates, M. C., Glaser, D., Post, J., Cristescu, M. E., Fraser, D. J., & Derry, A. M. (2020). The relationship between eDNA particle concentration and organism abundance in nature is strengthened by allometric scaling. *BioRxiv*, 2020.01.18.908251.

<https://doi.org/10.1101/2020.01.18.908251>

Line 171-172: For another passive filter study see;

Kirtane, A., Atkinson, J. D., & Sassoubre, L. (2020). Design and Validation of Passive Environmental DNA Samplers Using Granular Activated Carbon and Montmorillonite Clay. *Environmental Science & Technology*. <https://doi.org/10.1021/acs.est.0c01863>

179-183: Could also be statistical error associated with capture efficiency of either method. Most ecological sampling methods under sample natural communities. You would need to test the saturation of the different methods with increased replication to assess this for the current study.

What alternative materials are you referring to?

186-199: I would be careful in promoting one or the other. The difference seems more random. Yields were not presented, but if you would like to include them in the results it would be a nice discussion point.

Line 202-204: There is still time invested in setting up the apparatus and retrieving the samples compared to active filtering. Could you elaborate on actual time saved between the methods?

Line 206-207: The flow rates and volumes are well known to influence concentration volumes (the authors even mention this in the introduction). Several studies (two provided earlier) also show this.

Line 216-217: Not sure what the point is here. Beta diversity is a way to assess intercommunity differences and has been assessed from eDNA data.

Line 219-221: This is not accurate. Several measures of diversity utilize abundances and metabarcoding data derived diversity is increasingly using read numbers to derived proportional differences or weighted communities.

Reviewer #2 (Remarks to the Author):

The Study by Bessey et al, introduces a passive sampling method for eDNA capture from marine water to identify fish species using metabarcoding. The authors compare the results with those from active sampling, i.e. filtration of water. This advancement is exciting as it may help overcome some of the challenges faced using filtration methods, primarily by increasing the number of replicates that can be collected. This is the first publication reporting use of passive samplers for collecting eDNA in marine systems, but has been reported by Kirtane et al, 2020 in freshwater environment. The study found the membranes with neutral charge was better at capturing eDNA than positively charged. At one of the study sites, the passive samplers outperformed the active

samplers in detecting fish species. The authors did not find significant increase in fish detection by increasing the time samplers were submerged. This is a valuable addition to the currently growing realm of optimizing eDNA capture and analysis.

The methods section needs significant additions where the experimental design is mentioned in detail. There should be more information on active sampling methods used in this study. While, the data on fish species detected using metabarcoding is interesting, I am also interested to know the overall DNA yield from the passive samplers compared to active samplers. It may also be interesting to see the qPCR data, which the authors mention in the methods section – but do not show the results. Does the DNA yield and the qPCR signal higher in passive samplers? Does it increase with time the samplers were deployed? These questions are important to address to have a wider acceptance of the new method.

Title: Is misleading with the word “enhances”. Active samplers performed better 50% of the times and a much lower number of active samples were collected. Consider revising

Abstract:

Mention the potential mechanisms by which the DNA is passively captured. Add what membranes were tested as passive samplers.

Main

Introduction

The introduction needs to mention the research questions being addressed in the study. The introduction includes a rationale for why passive sampling may be useful. This should be elaborated and extended to explain the rationale as to why the specific submerged membranes were chosen in this study. This could include some explanation of the intended mechanisms (eg: adsorption, electrostatic binding, sieving, etc) of eDNA capture for the passive samplers. Overall, there needs to be more background for why these two membrane types were chosen for the study.

68: What materials were the passive sampling membranes made up of? The membrane material is an important component of the study and should be mentioned earlier in the paper.

Results

78: What two types of membranes?

80: How long were the membranes submerged for before retrieval? How many membranes were deployed? How were the membranes secured and deployed in the water? I see the information in Fig 1, but should also be mentioned in the text.

82-90: This reads more like a methods section than a results section. Consider revising.

100: Interesting result. What does this convey about the mechanism of eDNA capture? Maybe address in the discussion section.

102: If the sampling time did not affect the species detection, how do you know the eDNA was passively sampled? Passive sampling should lead to increased eDNA capture until saturation. If the membranes get saturated very fast, are they still sampling passively? Can you show the qPCR data or DNA yield from sampler over the time period?

119-124: mention this in the methods section. Also, total of active 9 samples were collected from Ashmore, how many from Dow Island?

127-141: So total of 64 taxa were detected from 68 passive samplers, and 84 from 9 active samplers at Ashmore. And 49 taxa were detected from 78 passive samples, and 40 from 9 active samplers at Dow. How much of this variation could be attributed to sampling effort?

Figure 2: This is a really good figure. How do you interpret the clustering of points for active filtration at Ashmore while they are quite scattered at Dow ?

131-132: Please elaborate further on how species similar species composition in active filtration reflects a greater number of species identified per sample. I am not able to follow this argument.

149-151: How was the variation for passive samples that were deployed for 4 hours vs 36 hours ? What does that tell about the utility of passive samplers, and how long they should be deployed in a given water matrix?

Discussion

156: Sentence needs grammatical revision. Not sure what you are trying to say.

158 – 160: This assumption needs more evidence to back it up. The detection rate is not only dependent on abundance but numerous other factors like primer bias, size of the organism,

metabolism, shedding rate, activity, and lots more. As of now, it has been well understood that reliable measures of species abundance or population cannot be made using eDNA metabarcoding.
163: Talk about how your approach is different from previously published work on eDNA passive sampling.

Kirtane, A., Atkinson, J. D., & Sassoubre, L. M. (2020). Design and Validation of Passive Environmental DNA Samplers (PEDS) using Granular Activated Carbon (GAC) and Montmorillonite Clay (MC). *Environmental Science & Technology*.

165- mention the two materials used in the passive samplers again here

166-168 – What are the new questions that can be answered using the passive sampling that were not possible using filtration. Might want to add a couple of specific examples.

177-179: Add figure reference at the end of this sentence.

179: What was the DNA yield of the passive samples compared to active samplers?

183: What alternative materials? Give some suggestions based on your results.

186: Again, remind the reader which two materials were evaluated at the beginning of this paragraph.

188: figure reference at the end of this sentence.

192-194: The references for this sentence all use active filtration. Can you be certain the same properties leading to higher yields in active filtration will also provide higher yields in passive filtration? The mechanism of eDNA capture in both methods may be completely different. Consider rewording the sentence.

206-207: And what kinds of questions would those be? Give specific examples if possible.

209: This is the first time authors have mentioned how the filters were deployed. And to my understanding the only place where this is mentioned. A detailed paragraph reporting how the membranes were deployed is required in the methods section.

229: The methods section seems to be missing a lot of information especially with the outline of the study design. While figure 1 shows the experimental setup for passive samplers, it should be accompanied by text explaining the rationale behind the time exposure of the membranes. Why were 4, 8, 12 and 24 hours chosen? Based on preliminary studies? Do the passive samplers saturate after 24 hours? One of the biggest strengths of passive sampling is collecting data over a period of time, instead of a single snapshot which could overcome the variability of the eDNA. Second, there is very little information on the methods used for active sampling. Did you use the same filters for active and passive sampling? What was the pore size? What volume was filtered per sample? Etc.

273: Why are the qPCR results not mentioned in the paper? Was the CT value of the passive samplers consistently lower than that of active filters further supporting the metabarcoding results? Maybe the authors could also include the DNA yield data (ng/ul) using Nanodrop or Qubit to check whether the passive samplers had a greater DNA yield.

Reviewer #3 (Remarks to the Author):

General comments

I have reviewed the manuscript 'Passive eDNA collection enhances aquatic biodiversity analysis'. This is a novel study evaluating an alternative strategy for aqueous eDNA capture to active filtration in both temperate and tropical ecosystems. I believe this work will drastically change the face of eDNA research and move the field forward in terms of the questions that can be addressed using both targeted and metabarcoding approaches. The study nicely shows that passive eDNA collection using charged and non-charged filter membranes can detect fish biodiversity, with non-charged membranes in particular achieving comparable or better detection

than active filtration. The experimental design, sampling strategy, and inferences are sound, and I commend the authors for a well-written manuscript and carefully designed study. However, I would like the authors to either restructure the manuscript to have clearer separation of Methods and Results, or add details to the Methods that are included in the Results but missing from the Methods. I would like the authors to clarify aspects of their methodology, but otherwise I have only minor comments to suggest. I have detailed these in the specific comments to the authors below.

Specific comments

Line 44: Change 'on both land and in the water' to ', both on land and in the water'.

Lines 78-90: The bulk of this subsection is Methods, not Results. Perhaps the authors are trying to summarise their methods in the Results as I note the Methods section is online only. If this is the case, there is methodological information contained in the Results that is missing from the Methods (sampling, eDNA capture) and should be provided there even if it creates some repetition.

Line 90: How many filters from active filtration contained fish taxa?

Lines 93-97: These sentences are Methods and not Results. I suggest created a new section in the Methods, titled 'eDNA sampling', 'eDNA capture' or 'Filter deployment', which includes the detail from Lines 78-81 and Lines 93-97. I would then remove these sentences from the Results.

Lines 103-105: Again, these sentences are Methods and not Results. I would put this information in a new section in the Methods.

Lines 111-124: This section is entirely Methods, not Results.

Line 115: How were membranes lost during retrieval? Did they fall off the aquaculture frame or dropped while handling with tweezers? The former may be an important consideration for people wishing to use passive eDNA filtration.

Lines 116-117: I'm assuming that the 5 day period means that 24 membranes were deployed for 3 days then another size were deployed for 34 hours separately, but the reasons for this need to be made clearer.

Lines 123-124: Does this mean the authors filtered at 08:00 on Day 1, 12:00 on Day 2, and 16:00 on Day 3, or at all three time intervals on each day at Daw Island? Some clarification needed as Line 146 says no replication of time points was achieved for Daw Island.

Line 161: Insert 'and filtration' after 'sampling'.

Lines 230-248: More details on sampling, eDNA capture, and contamination mitigation could be provided. A new section titled 'eDNA sampling', 'eDNA capture' or 'Filter deployment' could contain information currently given in the Results, for example, the material and pore size of the positively and non-charged membranes used for passive filtration. Additionally, the methods for active water filtration should be provided, i.e. sampling container, volume, filter material and pore size. For contamination mitigation, how was the pearl oyster aquaculture frame sterilised before use? Was it sterilised each time after filters were removed and before new filters were added? Did the positively and non-charged membranes come pre-sterilised or did the authors sterilise them before use? How were sampling containers for active filtration sterilised?

Line 247: Insert 'water' after 'deionized'.

Line 258: To me, the use of PCR duplicates is the biggest weakness in the study. Do the authors have any evidence to support that they will effectively recover the majority of biodiversity present, including rare species, with just two PCR replicates? Did they conduct any occupancy modelling to estimate detection probability?

Lines 259-261: Have these primers been evaluated in silico and in vitro for the study systems, either in the present study or elsewhere? A brief summary of their taxonomic coverage and resolution would be informative.

Line 324: Were the authors really using Bray-Curtis dissimilarity? It has been my understanding that Bray-Curtis dissimilarity is only appropriate for abundance data, and Sorensen index should be used to account for abundance when working with binary presence/absence datasets. This because Bray-Curtis dissimilarity applied to a binary presence/absence dataset becomes very similar to Jaccard dissimilarity.

Lines 325-327: Before or after applying PERMANOVA, did the authors test for homogeneity of multivariate dispersions (MVDISP) using the anova() or permutest() functions in vegan? This test is important to distinguish whether the differences observed between groups in nMDS are in fact due to community dissimilarity or uneven dispersions (variance) in one or more groups. Some would argue that PERMANOVA should not be performed if there is significant MVDISP because this violates one of the key assumptions of PERMANOVA. I would simply like to see the MVDISP results reported alongside those of the PERMANOVA so that readers can draw their own conclusions about the data.

Line 476 and 502: Change 'permutation MANOVAs' to 'PERMANOVAs' as this is how they have been referred to throughout the text, unless 'PERMANOVA' should be 'permutation MANOVAs' on Line 328. What were the R-squared and F values for each PERMANOVA test? I suggest including these in the tables in Figure 2 as well.

Line 482 and 509: Change 'Figure S1' to 'Figure S2'.

Lines 489-490: I would include the same detail on the coloured dots in the legend for Table 2 as was given in the legend for Table 1.

Please note all line numbers indicated below correspond to our revised submission in 'No Markup' view of 'Track Changes'.

Referee expertise:

Referee #1: eDNA

Referee #2: eDNA passive filtering

Referee #3: eDNA metabarcoding, environmental monitoring

Reviewers' comments:

Reviewer #1 (Remarks to the Author):

Overview

The authors presents a study that utilizes passive filter collection of environmental DNA and show that the results are comparable to active filtering methods. While the study is relevant to eDNA research, many of the statements regarding the differences between methodologies and the reason for using passive filtration could be improved and further supported throughout the manuscript in order to convince the reader that the methodology is a valid alternative to their current methods. Major considerations should be made to the structure of the introduction and discussion (comments below).

Abstract

Line 31: As compared to what?

This has been clarified as "passive eDNA collection can detect fish as effectively as active eDNA filtration methods in temperate systems and can also provide similar estimates of total fish biodiversity." (lines 30-31)

Both passive and active filtration require water to pass through a membrane. There are pros and cons with either methods, such as losing the passive membrane prior to retrieval or potential contamination from placing the filter directly in the sampling environment. Observational wise I would expect similar results (provided that the methods allowed for sufficient material collection), but I would be more concerned with contamination via the passive filtration.

The reviewer identifies two potential challenges with passive eDNA collection: membrane loss prior to retrieval and potential contamination. While membrane loss was not an issue in our study, it will be important for any passive eDNA collection device to firmly secure the membranes in place to prevent loss. We address this in the Discussion ... "Passive eDNA collection does, however, require anchoring and ensuring membranes are secured in place. We used a pearl oyster aquaculture frame with mesh pockets, which illustrates how low-cost options for secure deployment can be easily imagined." (lines 211-213)

Contamination is something that we have thought very extensively about. Placing a membrane in situ enables DNA to be collected from the surrounding environment. Provided membranes are responsibly handled upon retrieval (sterile gloves, tweezers and storage packaging), as is required for any eDNA study that collects water, sediment or recruitment plates, we see no additional avenue for contamination.

Introduction

Line 37 – 40 Thomese and willerslev proposed the orgin of eDNA. There have studies since that have assessed eDNA orgin and size that provide an updated view on eDNA state

Jo, T., Arimoto, M., Murakami, H., Masuda, R., & Minamoto, T. (2019). Particle Size Distribution of Environmental DNA from the Nuclei of Marine Fish. *Environmental Science & Technology*, 53(16), 9947–9956. <https://doi.org/10.1021/acs.est.9b02833>

Moushomi, R., Wilgar, G., Carvalho, G., Creer, S., & Seymour, M. (2019). Environmental DNA size sorting and degradation experiment indicates the state of *Daphnia magna* mitochondrial and nuclear eDNA is subcellular. *Scientific Reports*, 9(1). <https://doi.org/10.1038/s41598-019-48984-7>

Wilcox, T. M., McKelvey, K. S., Young, M. K., Lowe, W. H., & Schwartz, M. K. (2015). Environmental DNA particle size distribution from Brook Trout (*Salvelinus fontinalis*). *Conservation Genetics Resources*, 7(3), 639–641. <https://doi.org/10.1007/s12686-015-0465-z>

We have modified our sentence to reflect the reviewer's comments.

"Macro-organisms can shed their DNA into the air, soil, and water through many means, including faeces, sloughed tissue cells, or as gametes, which can be collected as intra- or extra-cellular particles^{2,3}. These DNA particles are then extracted, amplified using primer sets designed to target specific taxa, sequenced, and compared to a reference database of known DNA sequences to determine taxonomic identities⁴." (lines 37-41)

We have also included the fish related references suggested by the reviewer.

Line 40: Environmental DNA is not a technique, but rather encompasses a field of research
Deiner, K., Bik, H. M., Mächler, E., Seymour, M., Lacoursière-Roussel, A., Altermatt, F., Creer, S., Bista, I., Lodge, D. M., de Vere, N., Pfrender, M. E., & Bernatchez, L. (2017). Environmental DNA metabarcoding: Transforming how we survey animal and plant communities. *Molecular Ecology*. <https://doi.org/10.1111/mec.14350>

Bohmann, K., Evans, A., Gilbert, M. T. P., Carvalho, G. R., Creer, S., Knapp, M., Yu, D. W., & de Bruyn, M. (2014). Environmental DNA for wildlife biology and biodiversity monitoring. *Trends in Ecology & Evolution*, 29(6), 358–367. <https://doi.org/http://dx.doi.org/10.1016/j.tree.2014.04.003>

We have modified our sentences to explicitly describe the molecular techniques of extracting, amplifying and sequencing collected eDNA particles. We have also changed our wording to 'environmental DNA metabarcoding' to clarify any confusion for the reader.

"Environmental DNA (eDNA) metabarcoding is a sensitive and broadly applicable tool for biodiversity research and environmental monitoring that is non-invasive and promises to be both cost- and time-effective¹." (lines 35-37)

Line 41-42: This line needs support. See:

Seymour, M., Edwards, F. K., Cosby, B. J., Kelly, M. G., de Bruyn, M., Carvalho, G. R., & Creer, S. (2020). Executing multi-taxa eDNA ecological assessment via traditional metrics and interactive networks. *Science of The Total Environment*, 729, 138801.

<https://doi.org/10.1016/j.scitotenv.2020.138801>

Thank you for this reference, which we now used in support of our claims and included as #5 in our References. (lines 415-417)

Line 46-51: This section could be revise as several of the statements could be increased in accuracy.

E.g., the use of a pumping mechanisms is not universal for active filtration. For examples see:

Deiner, K., Walser, J.-C., Mächler, E., & Altermatt, F. (2015). Choice of capture and extraction methods affect detection of freshwater biodiversity from environmental DNA. *Biological Conservation*, 183, 53–63. <https://doi.org/http://dx.doi.org/10.1016/j.biocon.2014.11.018>

Deiner, K., Lopez, J., Bourne, S., Holman, L., Seymour, M., Grey, E. K., Lacoursière, A., Li, Y., Renshaw, M. A., Pfrender, M. E., Rius, M., Bernatchez, L., & Lodge, D. M. (2018). Optimising the detection of marine taxonomic richness using environmental DNA metabarcoding: the effects of filter material, pore size and extraction method. *Metabarcoding and Metagenomics*, 2.

<https://doi.org/10.3897/mbmg.2.28963>

We have modified this section to include the additional DNA concentration methods of centrifugation and precipitation. We use the suggested Deiner et al. 2015 paper as supporting evidence and is included as #12 in our References. (lines 435-437)

“Although centrifugation and precipitation can be used to concentrate eDNA from small water volumes¹², filtration is more often used because it can accommodate a larger water volume which improves detection of rare species¹¹.” (lines 49-51)

We have also tempered our language to indicated that a pumping mechanism is typical rather than universal.

“Despite the variability in strategies, the use of a pumping mechanism to achieve active filtration is typical¹⁰.” (lines 54-55)

Line 53-57: These statements are rather vague and could use more support. See Seymour, M., Durance, I., Cosby, B. J., Ransom-Jones, E., Deiner, K., Ormerod, S. J., Colbourne, J. K., Wilgar, G., Carvalho, G. R., de Bruyn, M., Edwards, F., Emmett, B. A., Bik, H. M., & Creer, S. (2018). Acidity promotes degradation of multi-species environmental DNA in lotic mesocosms.

Communications Biology, 1(1). <https://doi.org/10.1038/s42003-017-0005-3>

Deiner, K., & Altermatt, F. (2014). Transport Distance of Invertebrate Environmental DNA in a Natural River. PLoS ONE, 9(2), e88786. <http://dx.doi.org/10.1371/journal.pone.0088786>

We have now included examples which use the provided references to further support our statements.

“For example, acidic environments are known to accelerate the degradation of eDNA¹⁹, while cool, flowing water (i.e. river) systems may transport eDNA up to 10km from the source²⁰.” (lines 61-63)

Line 58: Most studies do not filter 20L and studies have also shown that 1-2L is sufficient.

Mächler, E., Deiner, K., Spahn, F., & Altermatt, F. (2016). Fishing in the Water: Effect of Sampled Water Volume on Environmental DNA-Based Detection of Macroinvertebrates. Environmental Science & Technology, 50(1), 305–312. <https://doi.org/10.1021/acs.est.5b04188>

We now include a more balanced approach as suggested by the reviewer, where we indicate that some studies may require as little as 1L of water.

“Although some studies indicate that sampling small water volumes (such as 1 L) can adequately detect macro-organisms of interest in some systems²¹, other studies suggest at least 20 L of water per site, or more, must be sampled before species accumulation curves approach an asymptote for diversity^{9,22,23}.” (lines 63-67)

Aims and objective should be clarified and hypotheses better supported via the introduction

We now include our hypothesis and rationale for trialling two membrane types in the Introduction.

“We hypothesized that both positively charge nylon and non-charged cellulose ester membranes can collect eDNA when placed in the water column through electrostatic attraction or entrapment, respectively.” (lines 78-80)

Methods

Lines 231-241: A lot of information is missing or is out of order. How many samples? Replications? Filter sizes? Length of passive filtration (particularly since you argue that the method saves time)? Active filtration equipment used? Were the samples preserved in any way or were they dried? What material was the sampling equipment? How was the equipment sterilized prior to each sampling event?

As suggested by all reviewers, we have reorganized both the Methods and Results sections to have clearer separation and to contain all the requested missing information. Please see our revised Methods sections entitled ‘Passive eDNA Collection’ and ‘Active eDNA Collection’ to answer the proposed questions. (lines 263-288)

Line 253: Did you manage to get sufficient DNA concentrations with 200 uL elutions? Many studies use lower elution volumes given the low concentration of eDNA in their extracts

As suggested by Reviewer #2, we now report mean Cq values from our qPCR duplicates by treatment and treatment x submersion duration (Fig.S1) as an indication of initial DNA copy number. These data are reported in our results section. (lines 107-112; 122-125)

Results

Line 78-80: How many samples and what was the interval. Don't make the reader go digging through figures to guess what the study design was.

Our substantial revisions of the Methods and Results has rectified this issue. "To examine whether increased submersion time of membranes led to increased detection, we retrieved triplicate membranes after four, eight, 12, and 24 hours of deployment (Fig.1b). This design was deployed at both our Ashmore Reef and Daw Island sites (Fig. 1c). Membranes were deployed over a three day period (June 17 to 20, 2019) at Ashmore Reef (3 days x 24 membranes – 4 membranes were lost during retrieval due to handling error; n = 68 membranes) and five days (Jan 29 to Feb 2, 2019) at Daw Island (3 days x 24 membranes + 6 membranes deployed for 34 hours; n = 78 membranes). Time allowed for the addition of a 34 hour deployment trial at Daw Island." (lines 272-279)

Line 84-86: Consider an alternative statement than choosing to analyze fish because they are popular

We have revised this statement in the methods section entitled 'DNA metabarcoding amplification from eDNA'. "We chose to analyse fish eDNA because this is one of the most common assessments made in aquatic eDNA metabarcoding studies^{6,8,9,10,11} and is associated with a substantial reference database for taxonomic identification." (lines 303-306)

Line 103-104: More details pertaining to the sampling apparatus construction and usage should be provided. How were the filters kept in place while they collected material? Was there any consideration for tidal or current influences?

As suggested by all reviewers, we have reorganized both the Methods and Results sections to have clearer separation and to contain all the requested missing information. As indicate now on lines 212 and 265, our membrane filters were held in place by the mesh pockets of the pearl frame, regardless of differences in tidal flow or current.

Line 121: This pump is a benchtop model. Were the samples transported to a lab prior to filtering?

We now clarify where the samples were filtered in a laboratory setting onboard the vessel. "Nine 1 L surface water samples were collected in sterile 1 L containers at both sites and filtered over nine non-charged cellulose membranes (47 mm diameter, 0.45 µm pore size) using a peristaltic Sentino® Microbiology Pump in a laboratory setting aboard the vessel." (lines 282-285)

Line 132-134: This belongs in the discussion

This sentence has been reworded to more appropriately report results. "Species composition was significantly different between all treatments (Fig. 2b and c for statistics; all p-values < 0.01), although the most abundant fish inhabiting Ashmore Reef²⁷, such as *Acanthurus triostegus*, and *Halichoeres trimaculatus* (Table 1), were detected by all collection methods." (lines 133-136)

Line 144-151: This belongs in the discussion as currently written

This section has been re-written to make it appropriate for the results section. "We found actively filtered samples displayed less variation in fish community at Ashmore Reef than Daw Island (Fig. 2b, grey dashed lines). We investigated this further and found that samples showed similarity by the time (Fig. S2) and day (Fig. S3) they were collected." (lines 147-149)

Discussion

Line 158-161: This is not supported presently, see

Iliana, B., R., C. G., Min, T., Kerry, W., Xin, Z., Mehrdad, H., Shadi, S., Mathew, S., David, B., Shanlin, L., Martin, C., & Simon, C. (2018). Performance of amplicon and shotgun sequencing for accurate biomass estimation in invertebrate community samples. *Molecular Ecology Resources*, 0(0).

<https://doi.org/10.1111/1755-0998.12888>

Yates, M. C., Glaser, D., Post, J., Cristescu, M. E., Fraser, D. J., & Derry, A. M. (2020). The relationship between eDNA particle concentration and organism abundance in nature is strengthened by allometric scaling. *BioRxiv*, 2020.01.18.908251. <https://doi.org/10.1101/2020.01.18.908251>

We have chosen to focus our discussion on the merits of using eDNA metabarcoding for relative rather than absolute abundance. Our point, which we have expanded upon and emphasise in lines 225-236 of the Discussion, is that differences in relative detection rates between sites can be measured once there are enough samples for frequency of occurrence analysis to be powerful enough to detect differences. This has been described previously in a metabarcoding context (Willerslev et al., 2014; Deagle et al., 2019) and is well established for any ecological method with differential detection rates.

Line 171-172: For another passive filter study see;

Kirtane, A., Atkinson, J. D., & Sassoubre, L. (2020). Design and Validation of Passive Environmental DNA Samplers Using Granular Activated Carbon and Montmorillonite Clay. *Environmental Science & Technology*. <https://doi.org/10.1021/acs.est.0c01863>

We became aware of Kirtane et al.'s exciting and similar study after our manuscript was submitted for publication and still under review. We now refer to their manuscript and talk about how our approach differs.

“Our results provide compelling evidence that eDNA can be passively collected from marine waters with minimal equipment and without using granular materials³¹, which require additional handling in the laboratory.” (lines 161-163)

“More recently, in freshwater microcosm and field experiments, granular activated carbon was shown to capture an order of magnitude more eDNA than montmorillonite clay³¹.” (lines 194-196)

179-183: Could also be statistical error associated with capture efficiency of either method. Most ecological sampling methods under sample natural communities. You would need to test the saturation of the different methods with increased replication to assess this for the current study.

We are keen to conduct further studies to investigate the saturation points of membranes. The addition of mean C_q values in the results (new Fig. S1) indicate that actively filtering water in a high diversity system can lead to higher initial copies of eDNA.

What alternative materials are you referring to?

We now give specific examples of alternative materials, such as “activated carbon or clay³¹.”

186-199: I would be careful in promoting one or the other. The difference seems more random. Yields were not presented, but if you would like to include them in the results it would be a nice discussion point.

We have tempered our language, which now reads “This may suggest that some materials outperform others for passive eDNA collection, and that the collection of eDNA on passive membranes may not be dominated by electrostatic attraction of naked DNA molecules to the membrane, but by other mechanisms”. (lines 188-190)

Yields of eDNA are difficult to quantify directly because they are very dilute. We have quantified qPCR amplification potential as a proxy and added the results in an additional figure (Fig. S1).

Line 202-204: There is still time invested in setting up the apparatus and retrieving the samples compared to active filtering. Could you elaborate on actual time saved between the methods? We now provide a practical example for reference. “For example, in the same time it took us to collect and filter a 1 L water sample at our anchor site aboard the vessel, we could deploy and retrieve all 24 membranes using our pearl frame apparatus.” (lines 207-209)

Line 206-207: The flow rates and volumes are well known to influence concentration volumes (the authors even mention this in the introduction). Several studies (two provided earlier) also show this. We have modified to this sentence to read, “Although the volume of water passing over the membrane is unknown, flow meters could be used to provide a proxy if necessary for the question being addressed.” (lines 209-211)

Line 216-217: Not sure what the point is here. Beta diversity is a way to assess intercommunity differences and has been assessed from eDNA data. The point we are making, which others have also made, is that measures of beta diversity as an assessment of intercommunity differences are frequently biased in eDNA studies because the sampling methods limit biological replication. Under-sampling of communities inflates beta diversity measures, so passive eDNA sampling will lead to more realistic eDNA metabarcoding measures of beta diversity.

Line 219-221: This is not accurate. Several measures of diversity utilize abundances and metabarcoding data derived diversity is increasingly using read numbers to derived proportional differences or weighted communities. We have tempered our language and our sentence now reads “A more widely accepted approach is to treat detection of any amount of a taxon’s DNA as a “presence” and to collect enough biological samples to enable comparative frequency analysis between treatments^{1,29,39}.” (lines 222-224)

Reviewer #2 (Remarks to the Author):

The Study by Bessey et al, introduces a passive sampling method for eDNA capture from marine water to identify fish species using metabarcoding. The authors compare the results with those from active sampling, i.e. filtration of water. This advancement is exciting as it may help overcome some of the challenges faced using filtration methods, primarily by increasing the number of replicates that can be collected. This is the first publication reporting use of passive samplers for collecting eDNA in marine systems, but has been reported by Kirtane et al, 2020 in freshwater environment. The study found the membranes with neutral charge was better at capturing eDNA than positively charged. At one of the study sites, the passive samplers outperformed the active samplers in detecting fish species. The authors did not find significant increase in fish detection by increasing the time samplers were submerged. This is a valuable addition to the currently growing realm of optimizing eDNA capture and analysis.

The methods section needs significant additions where the experimental design is mentioned in detail. There should be more information on active sampling methods used in this study. We have reorganized the Methods and Results sections to have clearer separation, with more details now included in both the Methods and Results. We have created new sections in the Methods entitled ‘Passive eDNA Collection’ and ‘Active eDNA Collection’ and the requested additional details are outlined in our response to Reviewer #3’s comments. (lines 263-288)

While, the data on fish species detected using metabarcoding is interesting, I am also interested to know the overall DNA yield from the passive samplers compared to active samplers. It may also be interesting to see the qPCR data, which the authors mention in the methods section – but do not show the results. Does the DNA yield and the qPCR signal higher in passive samplers? Does it increase with time the samplers were deployed? These questions are important to address to have a wider acceptance of the new method.

This is an excellent suggestion. We now include an additional figure which plots the mean Cq values by treatment and treatment x submersion duration (Fig.S1). All associated details and statistics have been incorporated into our updated Results section. (lines 107-112; 122-125)

Title: Is misleading with the word “enhances”. Active samplers performed better 50% of the times and a much lower number of active samples were collected. Consider revising

The point of passive sampling is not that any one sample performs better than active filtration. It is that you can collect many more samples (>20 time more) by this approach and have greater levels of biological replication. This does enhance biodiversity assessment by increasing the range of questions that can be addressed. We have expanded on this further in the Discussion.

“Frequency-based semi-quantification is inherently constrained by sample size⁴¹. The ability to collect large numbers of samples (thereby increasing sample size) from each site makes analysis by frequency of occurrence of a DNA sequence between sites powerful. Collection of many samples from microhabitats within each site also allows for fine scale distribution mapping of species where presence is inferred by a DNA sequence. The far greater sampling rate allowed by passive eDNA sampling would also enable a greater temporal sampling density so that extent of residence of species might be inferred in systems with rapidly-mixing water, or fast degradation of eDNA signal⁴¹. Finally, gamma biodiversity estimation will be improved through passive eDNA sampling of a range of microhabitats within a field site because each microenvironment produces a different alpha biodiversity⁹.” (lines 227-236)

We prefer the original title.

Abstract:

Mention the potential mechanisms by which the DNA is passively captured. Add what membranes were tested as passive samplers.

We now include the passive eDNA membrane materials in the Abstract. “Here we demonstrate how eDNA can be passively collected in both tropical and temperate marine systems by directly submerging membranes (positively charged nylon and non-charged cellulose ester) in the water column.” (lines 26-29)

Main

Introduction

The introduction needs to mention the research questions being addressed in the study. The introduction includes a rationale for why passive sampling may be useful. This should be elaborated and extended to explain the rationale as to why the specific submerged membranes were chosen in this study. This could include some explanation of the intended mechanisms (eg: adsorption, electrostatic binding, sieving, etc) of eDNA capture for the passive samplers. Overall, there needs to be more background for why these two membrane types were chosen for the study.

We now include our hypothesis and rationale for trialling two membrane types in the Introduction. “We hypothesized that both positively charge nylon and non-charged cellulose ester membranes can collect eDNA when placed in the water column through electrostatic attraction or entrapment, respectively.” (lines 78-80)

68: What materials were the passive sampling membranes made up of? The membrane material is an important component of the study and should be mentioned earlier in the paper.

We now include the passive eDNA membrane materials in the final paragraph of the Introduction. "By submerging secured membranes in the water column, we demonstrate the viability of passive eDNA collection. We hypothesized that both positively charge nylon and non-charged cellulose ester membranes can collect eDNA when placed in the water column through electrostatic attraction or entrapment, respectively." (lines 77-80)

Additional details are also included in the Methods section as "We trialled passive eDNA collection by submerging two membrane materials approximately one meter below the ocean surface in the mesh pockets of a pearl oyster aquaculture frame and collected them at specified intervals (Fig.1a). One membrane material was a positively charged electrostatic nylon (0.45 μm Biodyne™ B, 47mm), while the other was a non-charged, cellulose ester (0.45 μm Pall GN-6 Metrical®). We used a positively charged membrane because extracellular DNA is negatively charged due to the phosphate on the sugar phosphate backbone. All membranes were certified sterile upon purchase." (lines 264-270)

Results

78: What two types of membranes?

We now include the two membrane types in the Results section as "Two different membranes (charged nylon versus non-charged cellulose ester) were trialled to determine if the membrane material used during passive eDNA collection influenced fish detection." (lines 95-97)

Additional details are also included in the Methods section as detailed below. (lines 264-270)

80: How long were the membranes submerged for before retrieval? How many membranes were deployed? How were the membranes secured and deployed in the water? I see the information in Fig 1, but should also be mentioned in the text.

As suggestion, passive eDNA sampling is now detailed in a new section of the Methods entitled 'Passive eDNA Collection'. (lines 264-279)

" We trialled passive eDNA collection by submerging two membrane materials approximately one metre below the ocean surface in the mesh pockets of a pearl oyster aquaculture frame and collected them at specified intervals (Fig.1a). One membrane material was a positively charged electrostatic nylon (0.45 μm Biodyne™ B, 47mm), while the other was a non-charged, cellulose ester (0.45 μm Pall GN-6 Metrical®). We used a positively charged membrane because extracellular DNA is negatively charged due to the phosphate on the sugar phosphate backbone. All membranes were certified sterile upon purchase.

To examine whether increased submersion time of membranes led to increased detection, we retrieved triplicate membranes after four, eight, 12, and 24 hours of deployment (Fig.1b). This design was deployed at both our Ashmore Reef and Daw Island sites (Fig. 1c). Membranes were deployed over a three day period (June 17 to 20, 2019) at Ashmore Reef (3 days x 24 membranes – 4 membranes were lost during retrieval due to handling error; n = 68 membranes) and five days (Jan 29 to Feb 2, 2019) at Daw Island (3 days x 24 membranes + 6 membranes deployed for 34 hours; n = 78 membranes). Time allowed for the addition of a 34 hour deployment trial at Daw Island."

82-90: This reads more like a methods section than a results section. Consider revising.

We have revised both our Methods and Results sections to have clearer separation and to contain the requested additional details. Specific changes are detailed throughout, and in our response to Reviewer #3.

100: Interesting result. What does this convey about the mechanism of eDNA capture? Maybe address in the discussion section.

We suggest in the Discussion that our results indicate that passive eDNA collection is likely not dominated by electrostatic attraction of naked DNA molecules. “Nevertheless, our charged nylon membranes failed to detect fish on five occasions, whereas all non-charged cellulose membranes detected fish. This may suggest that some materials outperform others for passive eDNA collection, and that the collection of eDNA on passive membranes may not be dominated by electrostatic attraction of naked DNA molecules to the membrane, but by other mechanisms.” (lines 186-191)

102: If the sampling time did not affect the species detection, how do you know the eDNA was passively sampled? Passive sampling should lead to increased eDNA capture until saturation. If the membranes get saturated very fast, are they still sampling passively? Can you show the qPCR data or DNA yield from sampler over the time period?

The membranes do appear to be quickly saturating in amplifiable DNA. There is no active mechanism involved in the sampling process, so the term “passive” means that we did not pump water through the membrane. Even if there was larval settlement or some other process, “passive” is still the correct term.

Including the qPCR data is a great suggestion that has been incorporated into our revised Results section. We now include C_q values by treatment and treatment x submersion duration, which are visualized in Figure S1. (lines 107-112; 122-125)

119-124: mention this in the methods section. Also, total of active 9 samples were collected from Ashmore, how many from Dow Island?

As suggestion, active filtration sampling is now detailed in a new section of the Methods entitled ‘Active eDNA Collection’. “We collected water for active eDNA filtration to compare to our passive method. Nine 1 L surface water samples were collected in sterile 1 L containers at both sites and filtered over nine non-charged cellulose membranes (47 mm diameter, 0.45 μm pore size) using a peristaltic Sentino® Microbiology Pump in a laboratory setting aboard the vessel. At Ashmore reef, we actively filtered triplicate one litre water samples at 8:00, 12:00, and 16:00 on the final day of the experiment (June 20, 2019). At Daw Island, we actively filtered one litre samples at 08:00, 12:00, and 16:00 each day for all three days of the experiment (January 30 – February 1, 2019).” (lines 282-288)

127-141: So total of 64 taxa were detected from 68 passive samplers, and 84 from 9 active samplers at Ashmore. And 49 taxa were detected from 78 passive samples, and 40 from 9 active samplers at Dow. How much of this variation could be attributed to sampling effort?

As was also suggested by Reviewer #3, we have revised our statistical analysis. We now include the analysis of variance results, including R-squared and F values, in Figure 2c.

“Fish communities detected by each treatment were subjected to principal coordinate analysis (pcoa) using a Sorensen pair-wise dissimilarity matrix based on presence/absence of taxa (APE and BETAPART)^{47,48}. An analysis of variance on the dissimilarity matrix (adonis) was used to determine if treatment was a significant source of variation (VEGAN)⁴⁹ in the fish community composition. Pairwise comparisons (pairwise.perm.manova; RVAideMemoire)⁵⁰ were then used with Bonferroni adjustment to reveal which treatments were significantly different ($\alpha = 0.05$).” (lines 370-376)

Figure 2: This is a really good figure. How do you interpret the clustering of points for active filtration at Ashmore while they are quite scattered at Dow ?

Thank you. We suggest that the scattering in clustering of points is a result of sampling differences between sites, which is outlined in our Results section entitled "Collection Design for Active Filtration Can Influence Variance in Detection". "We found actively filtered samples displayed less variation in fish community at Ashmore Reef than Daw Island (Fig. 2b, grey dashed lines). We investigated this further and found that samples showed similarity by the time (Fig. S2) and day (Fig. S3) they were collected." (lines 147-149)

131-132: Please elaborate further on how species similar species composition in active filtration reflects a greater number of species identified per sample. I am not able to follow this argument. We now focus on describing the results and have removed the following statement, 'This likely reflects the greater number of species identified per sample by active filtration.'

Our revision states that "Species composition was significantly different between all treatments (Fig. 2b and c for statistics; all p-values < 0.01), although the most abundant fish inhabiting Ashmore Reef²⁷, such as *Acanthurus triostegus*, and *Halichoeres trimaculatus* (Table 1), were detected by all collection methods." (lines 133-136)

We also now include a better explanation of how species diversity per membrane type differs "We detected significantly more fish species per non-charged membrane at Ashmore Reef ($X^2 = 34.81$, $df = 1$, $p < 0.001$; Fig.2a left), where the mean number of taxa was more than three times that detected on charged membranes (10 versus 3, respectively). In contrast, at Daw Island, there was no significant difference in species detection between membrane materials ($X^2 = 2.21$, $df = 1$, $p < 0.14$; Fig.2b right). The mean number of fish detected per charged versus non-charged membrane was 8 and 11, respectively. Of the five membranes that yielded no fish eDNA, all were positively charged nylon (four from Ashmore Reef and one from Daw Island). For comparison, the mean number of fish taxa detected per non-charged actively filtered membrane (1 L of water) was 42 and 17 at Ashmore Reef and Daw Island, respectively." (lines 97-105)

Since each active membrane at Ashmore Reef contains a greater diversity of species (42 species / membrane) compared to 10 (non-charged) or 3 (charged) for passive membranes, there is greater species overlap within active membranes.

149-151: How was the variation for passive samples that were deployed for 4 hours vs 36 hours ? What does that tell about the utility of passive samplers, and how long they should be deployed in a given water matrix?

The newly included Fig. S1 of Cq mean helps to visualise the variation in initial DNA copy number by treatment and submersion time. In summary, Fig. S1 shows little effect of submersion time on variation. We suggest in the Discussion that "Further studies to determine the optimal membrane material to use for passive eDNA collection could help increase capture rate and improve detection efficiency, and would be aided by a mechanistic understanding of passive eDNA capture." (lines 196-198)

We also now include recently published information about passive eDNA sampling in freshwater systems as suggested by both Reviewer #1 and #2.

Discussion

156: Sentence needs grammatical revision. Not sure what you are trying to say.

We have revised this sentence to read "The promise of eDNA as a universal biomonitoring tool²⁸ has been realised in many respects for species detection and biodiversity studies, with publication rate growing rapidly^{6,9}." (lines 152-153)

158 – 160: This assumption needs more evidence to back it up. The detection rate is not only dependent on abundance but numerous other factors like primer bias, size of the organism, metabolism, shedding rate, activity, and lots more. As of now, it has been well understood that reliable measures of species abundance or population cannot be made using eDNA metabarcoding. Indeed, all these aforementioned factors can affect detection rate. However, they affect detection rate approximately equally in all samples. Our point, which we have expanded upon and emphasise in lines 227-236 of the Discussion, is that differences in relative detection rates between sites can be measured once there are enough samples for frequency of occurrence analysis to be powerful enough to detect differences. This has been described previously in a metabarcoding context (Willerslev et al., 2014; Deagle et al., 2019) and is well established for any ecological method with differential detection rates. We are not suggesting that this makes eDNA metabarcoding a tool for absolute quantification, but in combination with high levels of biological replication it is a tool for detecting relative differences in frequency among sites.

163: Talk about how your approach is different from previously published work on eDNA passive sampling.

Kirtane, A., Atkinson, J. D., & Sassoubre, L. M. (2020). Design and Validation of Passive Environmental DNA Samplers (PEDS) using Granular Activated Carbon (GAC) and Montmorillonite Clay (MC). *Environmental Science & Technology*.

We became aware of Kirtane et al.'s exciting and similar study after our manuscript was submitted for publication and still under review. We now refer to their manuscript and talk about how our approach differs. "Our results provide compelling evidence that eDNA can be passively collected from marine waters with minimal equipment and without using granular materials³¹, which require additional handling in the laboratory." (lines 161-163)

165- mention the two materials used in the passive samplers again here

We now mention the two materials used in the passive samples again.

"This was true for two alternative membrane materials (positively charged nylon and non-charged cellulose ester) and for the detection of fish taxa in both tropical and temperate environments." (lines 163-165)

166-168 – What are the new questions that can be answered using the passive sampling that were not possible using filtration. Might want to add a couple of specific examples.

As requested, we now include specific examples to support and expand upon this statement in our Discussion section entitled "Benefits and Limitations of Passive eDNA Collection."

"Frequency-based semi-quantification is inherently constrained by sample size⁴¹. The ability to collect large numbers of samples (thereby increasing sample size) from each site makes analysis by frequency of occurrence of a DNA sequence between sites powerful. Collection of many samples from microhabitats within each site also allows for fine scale distribution mapping of species where presence is inferred by a DNA sequence. The far greater sampling rate allowed by passive eDNA sampling would also enable a greater temporal sampling density so that extent of residence of species might be inferred in systems with rapidly-mixing water, or fast degradation of eDNA signal⁴¹. Finally, gamma biodiversity estimation will be improved through passive eDNA sampling of a range of microhabitats within a field site because each microenvironment produces a different alpha biodiversity⁹." (lines 227-236)

177-179: Add figure reference at the end of this sentence.

We now include a reference to Figure 2 in this sentence. “Although fewer fish taxa were detected on our submerged membranes at the tropical Ashmore reef site compared to active filtration (Fig. 2a), the richness of taxa detected by these methods was similar at the temperate site.” (lines 176-178)

179: What was the DNA yield of the passive samples compared to active samplers?

As an indication of initial DNA copy number, we now include an additional figure which plots the mean Cq values by treatment and treatment x submersion duration (Fig. S1). All associated details and statistics have been incorporated into our updated Results section. (lines 107-112; 122-125)

183: What alternative materials? Give some suggestions based on your results.

We now include suggestions of alternative materials such as presented by Kirtane et al. 2020. “or through the use of alternative materials like activated carbon or clay.” (line 182)

186: Again, remind the reader which two materials were evaluated at the beginning of this paragraph.

We now include a reminder of the different membrane materials used. “Both membrane materials (charged nylon and non-charged cellulose ester) enabled passive eDNA collection, regardless of submersion time.” (lines 185-186)

188: figure reference at the end of this sentence.

As we are referring to the number of charged nylon membranes that failed to detect fish, there is no associated figure to reference.

192-194: The references for this sentence all use active filtration. Can you be certain the same properties leading to higher yields in active filtration will also provide higher yields in passive filtration? The mechanism of eDNA capture in both methods may be completely different. Consider rewording the sentence.

During the review process, a new study investigating passive eDNA capture in freshwater stream systems was published online. The study investigated the use of granular activated carbon and montmorillonite clay as potential eDNA collection materials. This new information has been incorporated into our Discussion. “More recently, in microcosm and field experiments, granular activated carbon was shown to capture an order of magnitude more eDNA than montmorillonite clay³¹.” (lines 194-196)

206-207: And what kinds of questions would those be? Give specific examples if possible.

We now include specific examples of how passive eDNA collection will enable new research questions to be addressed. “The ability to collect large numbers of samples (thereby increasing sample size) from each site makes analysis by frequency of occurrence of a DNA sequence between sites powerful. Collection of many samples from microhabitats within each site also allows for fine scale distribution mapping of species where presence is inferred by a DNA sequence. The far greater sampling rate allowed by passive eDNA sampling would also enable a greater temporal sampling density so that extent of residence of species might be inferred in systems with rapidly-mixing water, or fast degradation of eDNA signal⁴¹. Finally, gamma biodiversity estimation will be improved through passive eDNA sampling of a range of microhabitats within a field site because each microenvironment produces a different alpha biodiversity⁹.” (lines 228-236)

209: This is the first time authors have mentioned how the filters were deployed. And to my understanding the only place where this is mentioned. A detailed paragraph reporting how the membranes were deployed is required in the methods section.

All relevant deployment information is now contained the new Methods sections entitled ‘Passive eDNA Collection’ and ‘Active eDNA Collection’. (lines 254-279)

“Passive eDNA Collection

We trialled passive eDNA collection by submerging two membrane materials approximately one metre below the ocean surface in the mesh pockets of a pearl oyster aquaculture frame and collected them at specified intervals (Fig.1a). One membrane material was a positively charged electrostatic nylon (0.45 µm Biodyne™ B, 47mm), while the other was a non-charged, cellulose ester (0.45 µm Pall GN-6 Metrice®). We used a positively charged membrane because extracellular DNA is negatively charged due to the phosphate on the sugar phosphate backbone. All membranes were certified sterile upon purchase.

To examine whether increased submersion time of membranes led to increased detection, we retrieved triplicate membranes after four, eight, 12, and 24 hours of deployment (Fig.1b). This design was deployed at both our Ashmore Reef and Daw Island sites (Fig. 1c). Membranes were deployed over a three day period (June 17 to 20, 2019) at Ashmore Reef (3 days x 24 membranes – 4 membranes were lost during retrieval due to handling error; n = 68 membranes) and five days (Jan 29 to Feb 2, 2019) at Daw Island (3 days x 24 membranes + 6 membranes deployed for 34 hours; n = 78 membranes). Time allowed for the addition of a 34 hour deployment trial at Daw Island.

Active eDNA Collection

We collected water for active eDNA filtration to compare to our passive method. Nine 1 L surface water samples were collected in sterile 1 L containers at both sites and filtered over nine non-charged cellulose membranes (47 mm diameter, 0.45 µm pore size) using a peristaltic Sentino® Microbiology Pump in a laboratory setting aboard the vessel. At Ashmore reef, we actively filtered triplicate one litre water samples at 8:00, 12:00, and 16:00 on the final day of the experiment (June 20, 2019). At Daw Island, we actively filtered one litre samples at 08:00, 12:00, and 16:00 each day for all three days of the experiment (January 30 – February 1, 2019).”

229: The methods section seems to be missing a lot of information especially with the outline of the study design. While figure 1 shows the experimental setup for passive samplers, it should be accompanied by text explaining the rationale behind the time exposure of the membranes. Why were 4, 8, 12 and 24 hours chosen? Based on preliminary studies? Do the passive samplers saturate after 24 hours? One of the biggest strengths of passive sampling is collecting data over a period of time, instead of a single snapshot which could overcome the variability of the eDNA. Second, there is very little information on the methods used for active sampling. Did you use the same filters for active and passive sampling? What was the pore size? What volume was filtered per sample? Etc. We have reorganized the Methods and Results sections to have clearer separation, with more details now included in both the Methods and Results. All requested information is now contained the new Methods sections entitled ‘Passive eDNA Collection’ and ‘Active eDNA Collection’. (lines 254-279)

273: Why are the qPCR results not mentioned in the paper? Was the CT value of the passive samplers consistently lower than that of active filters further supporting the metabarcoding results? Maybe the authors could also include the DNA yield data (ng/ul) using Nanodrop or Qubit to check whether the passive samplers had a greater DNA yield.

We now include an additional figure which plots the mean Cq values by treatment and treatment x submersion duration (Fig.S1). All associated details and statistics have been incorporated into our updated Results section. (lines 107-112; 122-125)

Reviewer #3 (Remarks to the Author):

General comments

I have reviewed the manuscript ‘Passive eDNA collection enhances aquatic biodiversity analysis’. This

is a novel study evaluating an alternative strategy for aqueous eDNA capture to active filtration in both temperate and tropical ecosystems. I believe this work will drastically change the face of eDNA research and move the field forward in terms of the questions that can be addressed using both targeted and metabarcoding approaches. The study nicely shows that passive eDNA collection using charged and non-charged filter membranes can detect fish biodiversity, with non-charged membranes in particular achieving comparable or better detection than active filtration. The experimental design, sampling strategy, and inferences are sound, and I commend the authors for a well-written manuscript and carefully designed study. However, I would like the authors to either restructure the manuscript to have clearer separation of Methods and Results, or add details to the Methods that are included details in the Results but missing from the Methods. I would like the authors to clarify aspects of their methodology, but otherwise I have only minor comments to suggest. I have detailed these in the specific comments to the authors below.

We have reorganized the Methods and Results sections to have clearer separation, with more details now included in both the Methods and Results. These specific changes are detailed below.

Specific comments

Line 44: Change 'on both land and in the water' to ', both on land and in the water'.

We have changed 'on both land and in the water' to 'both on land and in the water' as suggested.

Lines 78-90: The bulk of this subsection is Methods, not Results. Perhaps the authors are trying to summarise their methods in the Results as I note the Methods section is online only. If this is the case, there is methodological information contained in the Results that is missing from the Methods (sampling, eDNA capture) and should be provided there even if it creates some repetition.

We have now moved methodological details to the 'Methods' section and retained only those details necessary to bring context to the results.

Line 90: How many filters from active filtration contained fish taxa?

The requested information is now included as "Fish taxa were detected from 141 of 146 passively deployed membranes (97%; 66/70 at Ashmore Reef and 77/78 at Daw Island) and on all 18 actively filtered membranes (100%; 9/9 at Ashmore Reef and 9/9 at Daw Island)." (lines 90-92)

Lines 93-97: These sentences are Methods and not Results. I suggest created a new section in the Methods, titled 'eDNA sampling', 'eDNA capture' or 'Filter deployment', which includes the detail from Lines 78-81 and Lines 93-97. I would then remove these sentences from the Results.

Lines 103-105: Again, these sentences are Methods and not Results. I would put this information in a new section in the Methods.

Lines 111-124: This section is entirely Methods, not Results.

As suggested, we have created new sections in the Methods entitled 'Passive eDNA Collection' and 'Active eDNA Collection' which now contain the methodological details. (lines 254-279)

The Results section has been re-written to only contain enough methodological detail to assist the reader with context. (lines 86-149)

Line 115: How were membranes lost during retrieval? Did they fall off the aquaculture frame or dropped while handling with tweezers? The former may be an important consideration for people wishing to use passive eDNA filtration.

We now include that "4 membranes were lost during retrieval due to handling error". (line 267)
No membranes were lost from the aquaculture frame.

Lines 116-117: I'm assuming that the 5 day period means that 24 membranes were deployed for 3 days then another size were deployed for 34 hours separately, but the reasons for this need to be made clearer.

We now include our rationale for conducting a longer time deployment.

“Membranes were deployed over a three day period (June 17 to 20, 2019) at Ashmore Reef (3 days x 24 membranes – 4 membranes were lost during retrieval due to handling error; n = 68 membranes) and five days (Jan 29 to Feb 2, 2019) at Daw Island (3 days x 24 membranes + 6 membranes deployed for 34 hours; n = 78 membranes). Time allowed for the addition of a 34 hour deployment trial at Daw Island.” (lines 265-270)

Lines 123-124: Does this mean the authors filtered at 08:00 on Day 1, 12:00 on Day 2, and 16:00 on Day 3, or at all three time intervals on each day at Daw Island? Some clarification needed as Line 146 says no replication of time points was achieved for Daw Island.

We now clarify that “At Daw Island, we actively filtered one litre samples at 08:00, 12:00, and 16:00 each day for all three days of the experiment (January 30 – February 1, 2019).” (lines 278-279)

Line 161: Insert ‘and filtration’ after ‘sampling’.

We now include ‘and filtration’ after ‘sampling’.

Lines 230-248: More details on sampling, eDNA capture, and contamination mitigation could be provided. A new section titled ‘eDNA sampling’, ‘eDNA capture’ or ‘Filter deployment’ could contain information currently given in the Results, for example, the material and pore size of the positively and non-charged membranes used for passive filtration. Additionally, the methods for active water filtration should be provided, i.e. sampling container, volume, filter material and pore size. For contamination mitigation, how was the pearl oyster aquaculture frame sterilised before use? Was it sterilised each time after filters were removed and before new filters were added? Did the positively and non-charged membranes come pre-sterilised or did the authors sterilise them before use? How were sampling containers for active filtration sterilised?

As suggested, we have created new sections in the Methods entitled ‘Passive eDNA Collection’ and ‘Active eDNA Collection’ which contain the requested details. The sampling container size is indicated as ‘sterile 1 L containers’, the volume is indicated as ‘Nine 1 L surface water samples’, the filter material is indicated as ‘positively charged electrostatic nylon (0.45 µm Biodyne™ B, 47mm)’ and ‘non-charged, cellulose ester (0.45 µm Pall GN-6 Metrice®)’ which includes pore size (0.45 µm). We now also specify how ‘Prior to use, all collection and deployment apparatus was sterilized by soaking in 10% bleach solution for at least 15 minutes and rinsed in deionized water. We also now include that ‘All membranes were certified sterile upon purchase.’ (lines 255-279)

Line 247: Insert ‘water’ after ‘deionized’.

We now include ‘water’ after ‘deionized’.

Line 258: To me, the use of PCR duplicates is the biggest weakness in the study. Do the authors have any evidence to support that they will effectively recover the majority of biodiversity present, including rare species, with just two PCR replicates? Did they conduct any occupancy modelling to estimate detection probability?

We agree that where eDNA metabarcoding is used as a survey tool, the goal is often to maximize detection probability. The limitations of DNA metabarcoding by any approach for alpha biodiversity measurement are well documented in recent literature (Zinger et al., 2019, Taberlet et al., 2018). The purpose of our study was to determine if passive eDNA sampling was a viable collection method, rather than maximise alpha biodiversity. We were directly comparing passive eDNA collection with filtering of water (as a more typical collection method) to acquire eDNA and evaluate their use as DNA metabarcoding substrates. The great advantage of passive aquatic eDNA sampling is that many more biological samples can be taken than is practical with water filtering. Whatever the number of PCR replicates used, or the sequencing depth decided upon, having more biological samples collected by passive aquatic eDNA sampling could potentially improve gamma biodiversity

measurement.

Lines 259-261: Have these primers been evaluated in silico and in vitro for the study systems, either in the present study or elsewhere? A brief summary of their taxonomic coverage and resolution would be informative.

To address this comment, we re-processed our sequence data following the OBITools tutorial (<https://pythonhosted.org/OBITools/wolves.html>). This entailed using 'ecoPCR' to simulate an in silico PCR using our 16S Fish primer assay. Our reference database was rebuilt on 27/08/2020 and will be provided online through the CSIRO data access portal. This is now included in the Methods as "This sequence processing directly follows the procedure described at <https://pythonhosted.org/OBITools/wolves.html>. Our reference database built in silico using our universal fish primer assay (on 27/08/2020) is provided." (lines 341-344)

Our re-analysis resulted in the detection of additional fish species and resulted in only minor differences to data interpretation.

Line 324: Were the authors really using Bray-Curtis dissimilarity? It has been my understanding that Bray-Curtis dissimilarity is only appropriate for abundance data, and Sorensen index should be used to account for abundance when working with binary presence/absence datasets. This because Bray-Curtis dissimilarity applied to a binary presence/absence dataset becomes very similar to Jaccard dissimilarity. This, and the following comment, are addressed together below.

Lines 325-327: Before or after applying PERMANOVA, did the authors test for homogeneity of multivariate dispersions (MVDISP) using the anova() or permutest() functions in vegan? This test is important to distinguish whether the differences observed between groups in nMDS are in fact due to community dissimilarity or uneven dispersions (variance) in one or more groups. Some would argue that PERMANOVA should not be performed if there is significant MVDISP because this violates one of the key assumptions of PERMANOVA. I would simply like to see the MVDISP results reported alongside those of the PERMANOVA so that readers can draw their own conclusions about the data.

Thank you for these suggestions. We have revised our statistical analysis.

"Fish communities detected by each treatment were subjected to principal coordinate analysis (pcoa) using a Sorensen pair-wise dissimilarity matrix based on presence/absence of taxa (APE and BETAPART)^{47,48}. An analysis of variance on the dissimilarity matrix (adonis) was used to determine if treatment was a significant source of variation (VEGAN)⁴⁹ in the fish community composition. Pairwise comparisons (pairwise.perm.manova; RVAideMemoire)⁵⁰ were then used with Bonferroni adjustment to reveal which treatments were significantly different (\$\alpha = 0.05\$ )." (lines 361-367)

Line 476 and 502: Change 'permutation MANOVAs' to 'PERMANOVAs' as this is how they have been referred to throughout the text, unless 'PERMANOVA' should be 'permutation MANOVAs' on Line 328. What were the R-squared and F values for each PERMANOVA test? I suggest including these in the tables in Figure 2 as well.

We now include the analysis of variance results, including R-squared and F values, in Figure 2 as suggested.

Line 482 and 509: Change 'Figure S1' to 'Figure S2'.

We've now corrected the labelling to 'Figure S2'.

Lines 489-490: I would include the same detail on the coloured dots in the legend for Table 2 as was given in the legend for Table 1.

We now include the same detail on the coloured dots in the legend for Table 2 as was given in the legend for Table 1.

REVIEWERS' COMMENTS:

Reviewer #1 (Remarks to the Author):

The authors have fully addressed my concerns and the manuscript is a very nice addition to the field. The novel and through test of passive filtration is greatly appreciated. A few minor suggestions below.

47: Write out the full name "Environmental DNA" to avoid starting the sentence with an acronym

74-76: Consider adding a link between active and passive filter to make the transition from the active filtration focus to the potential or actual benefits of passive filtration that your study is looking to investigate.

153-155: I don't quite follow this statement. There are numerous studies that have done just this.

160-163: This is perhaps the sentence you want to have as the first topic sentence in the discussion

174-175: Consider avoiding the word "some" and specifically stating what circumstances are being referred to give more weight to the statement.

178: Try to replace "this" with what is being reference to avoid any potential confusion.

198-200: Not sure what is meant by this statement. Consider clarifying.

Reviewer #2 (Remarks to the Author):

I have reviewed the revised manuscript and rebuttal letter from Bessey et al, and acknowledge that all the comments made during the last round of review were addressed by the authors. The methods section is now more organized, detailed and clear. The addition of Fig S1 is helpful for the reader conceptualize how eDNA may be captured by passive sampler membranes with different surface properties and different exposure times.

This study is a valuable contribution to advancing the use of passive sampling in eDNA studies.

Please note all line numbers indicated below correspond to our revised submission in 'No Markup' view of 'Track Changes'.

Dear Dr Bessey,

Your manuscript entitled "Passive eDNA collection enhances aquatic biodiversity analysis" has now been seen again by our referees, whose comments appear below. In light of their advice I am delighted to say that we are happy, in principle, to publish a suitably revised version in Communications Biology under the open access CC BY license (Creative Commons Attribution v4.0 International License).

We therefore invite you to revise your paper one last time to address the remaining concerns of our reviewers. At the same time we ask that you edit your manuscript to comply with our format requirements and to maximise the accessibility and therefore the impact of your work.

We hope to hear from you within two weeks. If you expect the process to take longer than one month, please let us know.

Congratulations on an excellent paper!

Best regards,

Luke Grinham, PhD
Associate Editor, Communications Biology
4 Crinan Street
London N1 9XW, UK
orcid.org/0000-0001-5583-8052
luke.grinham@nature.com

REVIEWERS' COMMENTS:

Reviewer #1 (Remarks to the Author):

The authors have fully addressed my concerns and the manuscript is a very nice addition to the field. The novel and through test of passive filtration is greatly appreciated. A few minor suggestions below.

47: Write out the full name "Environmental DNA" to avoid starting the sentence with an acronym

We now write out the full name "Environmental DNA" to avoid starting the sentence with an acronym. (line 47)

74-76: Consider adding a link between active and passive filter to make the transition from the active filtration focus to the potential or actual benefits of passive filtration that your study is looking to investigate.

Good idea. We now include an additional sentence to help transition the focus from active filtration to the benefits of passive filtration with the following sentence: "Low-cost, easily deployable alternatives, that do not require sophisticated equipment, and eliminate the need for time-consuming filtration, warrant investigation." (lines 74-76)

153-155: I don't quite follow this statement. There are numerous studies that have done just this.

Although positive correlations between eDNA metabarcoding data and fish abundance or biomass have been demonstrated, there is still controversy and uncertainty regarding their quantitative power among the scientific community and monitoring agencies (Fonseca 2018, Lab et al. 2019). We

believe it is important to reflect this and to point to the use of passively collected eDNA as potentially a partial solution. We have modified the relevant paragraph to read “The promise of eDNA as a universal biomonitoring tool²⁹ has been realised in many respects for species detection and biodiversity studies, with the publication rate growing rapidly^{6,9}. So far, however, the use of eDNA for the estimation of abundance and beta diversity has been more limited^{30,31}. It is possible to use the number of samples in which an individual species is detected as a proxy measure of relative abundance, provided there is a high degree of biological replication³². Yet, this level of replication is rarely achieved with conventional active eDNA sampling and filtration³³. Our passive eDNA collection method will allow for increased biological replication in the field, thereby permitting new types of ecological questions to be addressed³⁴. Passive eDNA sampling will improve beta diversity estimates by increasing the number of biological replicates taken per area. Likewise, relative abundance, prevalence or biomass estimation through frequency of occurrence metrics will be improved by collection of more samples.” (lines 161-172)

Fonseca VG (2018) Pitfalls in relative abundance estimation using eDNA metabarcoding. *Molecular Ecology Resources* 18(5): 923–926. <https://doi.org/10.1111/1755-0998.12902>

Lamb PD, Hunter E, Pinnegar JK, Creer S, Davies RG, Taylor MI (2019) How quantitative is metabarcoding: A meta-analytical approach. *Molecular Ecology* 28(2): 420–430. <https://doi.org/10.1111/mec.14920>

160-163: This is perhaps the sentence you want to have as the first topic sentence in the discussion. As suggested, we now use the following as the first topic sentences of the discussion. “We provide an alternative aquatic eDNA collection approach that does not require active filtration. Our results provide compelling evidence that eDNA can be passively collected from marine waters with minimal equipment and without using granular materials³¹, which require additional handling in the laboratory. This was true for two alternative membrane materials (positively charged nylon and non-charged cellulose ester) and for the detection of fish taxa in both tropical and temperate environments.” (lines 154-159)

174-175: Consider avoiding the word “some” and specifically stating what circumstances are being referred to give more weight to the statement.
In place of “some”, we now specify “in temperate ocean conditions”. (line 179)

178: Try to replace “this” with what is being referred to avoid any potential confusion.
As suggested we have changed “this” to “Differences in taxa detection” to avoid any potential confusion. (line 183)

198-200: Not sure what is meant by this statement. Consider clarifying.
We have re-written the sentences to read “Trialling additional materials, either those with high binding affinities or dense surface areas, could help identify ways to increase DNA capture rate and improve detection efficiency, which would be aided by a mechanistic understanding of passive eDNA capture. It is likely that much of what we term “eDNA” is DNA bound with other cellular components, so the properties of pure DNA (e.g. negatively charged backbone) may be less important in determining eDNA recovery rates than expected.” (lines 202-207)

Reviewer #2 (Remarks to the Author):

I have reviewed the revised manuscript and rebuttal letter from Bessey et al, and acknowledge that

all the comments made during the last round of review were addressed by the authors. The methods section is now more organized, detailed and clear. The addition of Fig S1 is helpful for the reader conceptualize how eDNA may be captured by passive sampler membranes with different surface properties and different exposure times.

This study is a valuable contribution to advancing the use of passive sampling in eDNA studies.

Thank you.